**Registered Report**

# Feature selection methods affect the performance of scRNA-seq data integration and querying

Luke Zappia [1,2], Sabrina Richter[1], Ciro Ramírez-Suástegui[1,3], Raphael Kfuri-Rubens [1,4,5], Larsen Vornholz [1], Weixu Wang[1], Oliver Dietrich [1,6], Amit Frishberg[1], Malte D. Luecken [1,7] & Fabian J. Theis [1,2,8] ✉

The availability of single-cell transcriptomics has allowed the construction of reference cell atlases, but their usefulness depends on the quality of dataset integration and the ability to map new samples. Previous benchmarks have compared integration methods and suggest that feature selection improves performance but have not explored how best to select features. Here, we benchmark feature selection methods for single-cell RNA sequencing integration using metrics beyond batch correction and preservation of biological variation to assess query mapping, label transfer and the detection of unseen populations. We reinforce common practice by showing that highly variable feature selection is effective for producing high-quality integrations and provide further guidance on the effect of the number of features selected, batch-aware feature selection, lineage-specific feature selection and integration and the interaction between feature selection and integration models. These results are informative for analysts working on large-scale tissue atlases, using atlases or integrating their own data to tackle specific biological questions.

Single-cell transcriptomics technologies are now accessible to many biological researchers. As the number of single-cell RNA sequencing (scRNA-seq) datasets has increased and analysis methods have improved, we are seeing a shift from exploratory experiments toward multi-sample datasets. This trend includes more designed experiments investigating specific phenomena or testing differences between conditions and larger efforts to catalog the cellular heterogeneity within tissues. More samples allow a deeper study of biology but present additional challenges including successful integration of samples to remove technical differences while conserving interesting biological variation. Good quality integration is especially critical for large-scale human atlas-building enterprises, where fully capturing tissue heterogeneity requires samples from a variety of individuals across locations, collected in different ways from different organ areas and profiled using a range of protocols or technologies[1].

Many computational scientists have tackled the integration problem and at least 250 tools for single-cell integration are now available[2]. Studies have evaluated the performance of some methods[3–6], leading

[1]Institute of Computational Biology, Computational Health Center, Helmholtz Munich, Neuherberg, Germany. [2]School of Computing, Information and Technology, Technical University of Munich, Munich, Germany. [3]Wellcome Sanger Institute, Wellcome Genome Campus, Hinxton, Cambridge, UK. [4]School of Medicine, Technical University of Munich, Munich, Germany. [5]Klinikum rechts der Isar, IIIrd Medical Department, Munich, Germany. [6]Helmholtz Institute for RNA-based Infection Research, Helmholtz Centre for Infection Research, Würzburg, Germany. [7]Institute of Lung Health & Immunity, Helmholtz Munich; Member of the German Center for Lung Research (DZL), Munich, Germany. [8]School of Life Sciences Weihenstephan, Technical University of Munich, Friesing, Germany. ✉e-mail: fabian.theis@helmholtz-munich.de

to a set of established metrics for assessing integration performance. While the methods have been compared, preprocessing steps that may affect integration have largely been overlooked. One step that has received some attention is feature selection, where benchmarks have shown that using highly variable genes generally leads to better integrations[3]; however, this study only considered one commonly used feature selection method. Unlike other analysis steps, such as clustering[7,8], the best feature selection approach for integration has not been assessed. Additional questions arise when considering how the integrated space is used as a reference to analyze further query samples. It is possible that selecting features could result in better integration of reference samples while at the same time leading to an integration model that is ignorant of alternative sources of biological variation relevant to understanding other samples.

This study assesses the impact of feature selection on integrating scRNA-seq samples and using the integrated reference to analyze query samples. We evaluate the performance of variants of over 20 feature selection methods using a range of metrics divided into five categories: batch effect removal, conservation of biological variation, quality of query to reference mapping, label transfer quality and ability to detect unseen populations (Extended Data Fig. 1). The results from our robust benchmarking pipeline (Extended Data Fig. 2) are informative for researchers integrating their own datasets or creating reference atlases, leading to better community resources and further biological insights.

The study was conducted in accordance with the registered, peer-reviewed protocol at https://doi.org/10.6084/m9.figshare.24995690.v1 (ref. 9). Except for pre-registered and approved pilot data, all analysis results reported in the paper were collected after the date of the registered protocol publication.

## Results

### Metric selection is critical for reliable benchmarking

For this study, we collected a wide variety of metrics covering different aspects of integration and query mapping. While measuring a broad range of factors is important, the behavior of many of these metrics has not been thoroughly characterized. This characterization is particularly important in our context as we use metrics developed to compare different integration approaches to instead assess the effect of feature selection methods. For this reason, we include a metric selection step to profile metrics and decide which to use for benchmarking. This step aims to select metrics that effectively measure performance, are not overly associated with technical factors and are nonredundant.

We performed the metric selection using random and highly variable (scanpy[10] implementation of a Seurat algorithm[11]) feature sets of different sizes for each dataset, performing integration and mapping, calculating metric scores and comparing the results (Fig. 1a). The observed range of scores was calculated using the random gene sets for each dataset–integration combination. We also used random sets to calculate the correlation between metrics and technical aspects of datasets (number of features, number of reference cells, number of reference labels and batches, number of query cells and number of query batches and unseen labels). We calculated the correlation between metric scores and the number of selected features using the highly variable feature sets as random feature sets do not have any inherent ordering (the first 100 features are no more informative than the next 100). An ideal metric would accurately measure what it is designed for, returning scores across its whole output range that are independent of technical features of the data and are orthogonal to other metrics in the study. Figure 1b shows a summary of the metric evaluation.

Using these results, we selected metrics to evaluate feature selection methods. We found that some metrics, such as batch average silhouette width (Batch ASW)[3] and k-nearest neighbors (kNN) correlation[12], showed little variation, even across a wide range of selected feature sets; however, this is not always easy to interpret. For example, the cell-type local inverse Simpson's index (cLISI)[13] metric has

a natural range of zero to the number of labels in the dataset, which are rescaled to be between zero and one, compressing the observed range so that even small differences can be informative. When considering the correlation of metrics with the number of selected features, we found that most metrics are positively correlated with the number of selected features, with a mean correlation of around 0.5. A few metrics (local structure[14] and kNN correlation) showed stronger and more consistent associations with the number of features. In contrast, the mapping metrics are generally negatively correlated. This relationship could be because smaller feature sets produce noisier integrations where cell populations are mixed. This scenario requires less-precise query mapping where mapping somewhere within the mixed population is sufficient to receive a high mapping score.

The effect of technical factors of datasets on metric scores is more difficult to interpret as we consider relatively few datasets here, and the factors are associated across datasets (a dataset with more cells typically has more batches and labels). We see that more complex datasets generally result in lower scores for all metrics (Extended Data Fig. 3). The exceptions to this are the Milo[15] and Uncertainty metrics. For Milo, it is difficult to say if the positive association between scores and technical factors is a general effect of having more data or an effect of individual features. In the case of the Uncertainty metric, it is likely that the classifier model used is not well calibrated and is less certain (giving higher scores) for more complex datasets regardless of any specific technical factor. Proper assessment of the effect of technical dataset features would require more datasets where each factor is varied independently, potentially through a simulation study.

Perhaps the most important consideration for metric selection is the correlation between metrics (Fig. 1b and Extended Data Fig. 3). We want metrics that measure different aspects of integration and query mapping and selecting several highly correlated metrics would bias our results in that direction. This effect is evident in the Integration (Bio) category where several metrics (adjusted Rand index (ARI), batch-balanced ARI (bARI)[16], normalized mutual information (NMI), batch-balanced NMI (bNMI)[16], cLISI, label average silhouette width (Label ASW)[3] and Local structure) are highly correlated with each other, prompting us to select only a subset of these. The classification metrics show even stronger correlations, with all metrics having similar scores. Here, we also selected a representative sample of metrics, but using only one or all metrics would have little effect on the results. The other consideration for metric correlations is the correlation between metric types. To aid interpretation, we want to be able to summarize these aspects individually, and correlations between opposing metric types make this difficult. This categorization is difficult for the case of the kBET metric[17], which is placed in the Integration (Batch) category but is also correlated with metrics that measure the conservation of biological variation. While this may be desirable for a single metric, including kBET in our study would confuse the signal between those categories. Another metric that stands out is graph connectivity[3], which was considered a batch correction metric by the original authors but is negatively correlated with other metrics in this category and positively correlated with Integration (Bio) metrics. We have kept this metric for the evaluation but include it in the Integration (Bio) category in all further analyses.

Based on this analysis we selected three Integration (Batch) metrics (batch principal-component regression (Batch PCR)[3], cell-specific mixing score (CMS) and integration local inverse Simpson's index (iLISI)[13], six Integration (Bio) metrics (isolated label ASW[3], isolated label F1 (ref. 3), bNMI, cLISI, local density factor difference (ldfDiff)[18] and graph connectivity), four mapping metrics (Cell distance[12], Label distance[12], mapping local inverse Simpson's index (mLISI)[12] and query local inverse Simpson's index (qLISI)[12]), three classification metrics (F1 (Macro), F1 (Micro) and F1 (Rarity)[19]) and three unseen population metrics (Milo, Unseen cell distance and Unseen label distance). Extended Data Table 1 gives our reasoning for excluding metrics.

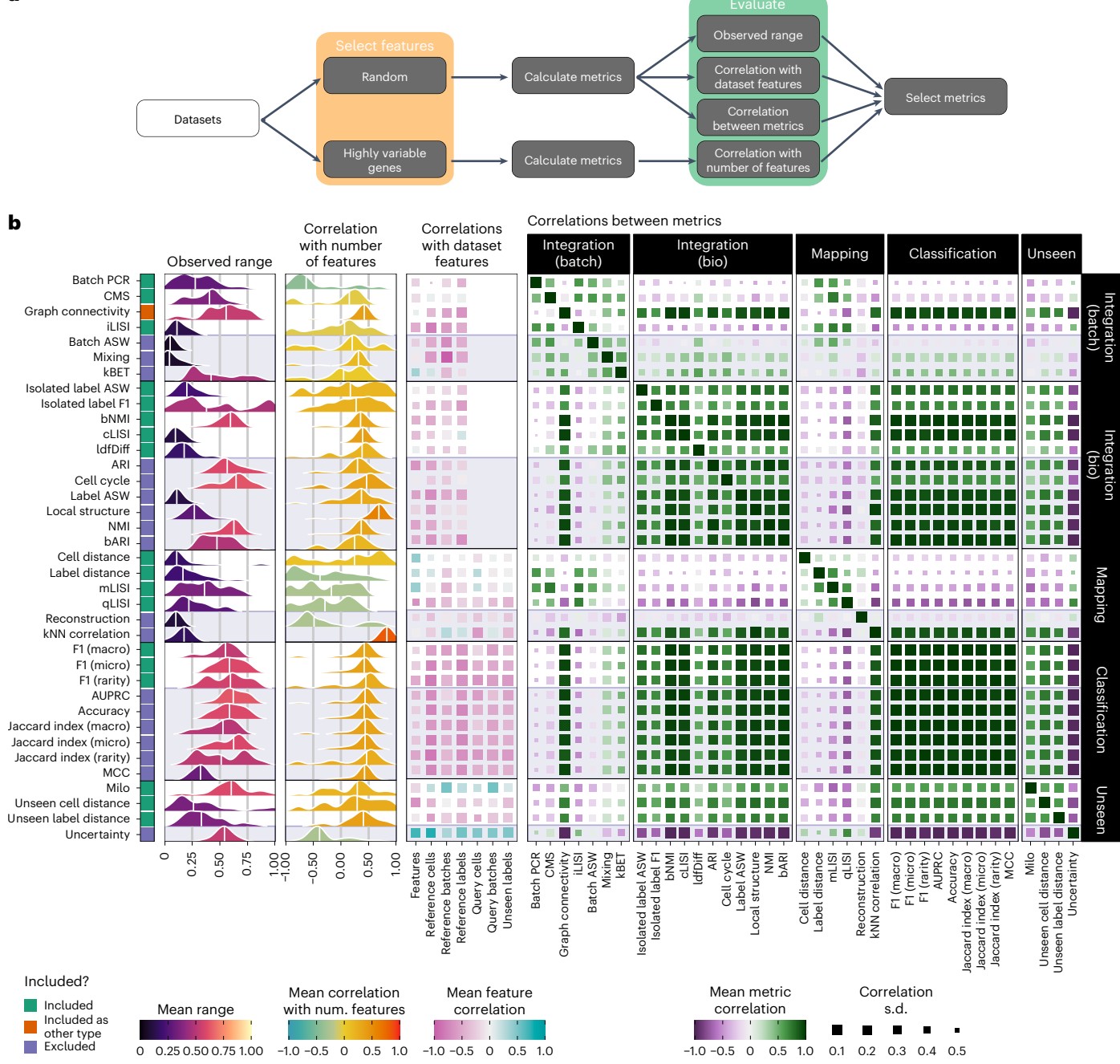

**Fig. 1 | Overview and results of the metric selection step. a**, Diagram of the metric selection workflow. **b**, Results of the metric selection step. Densities for the observed range and correlation with the number of features across datasets and integrations are shown for each metric. Colors indicate the mean value and vertical lines represent the median. The middle heatmap shows the mean correlation with technical dataset features (Extended Data Fig. 3a). Color indicates the mean correlation, and the size of squares is the s.d. (larger points are less variable). The heatmap on the right shows the mean correlation between metrics grouped by metric type (Extended Data Fig. 3b). The color bar on the left indicates which metrics were selected for the final benchmark. This indication is continued as shaded areas in the other plots.

**Using baselines to effectively scale and summarize metrics**

Individual metrics have different effective ranges and interact differently with datasets. To summarize and compare metric scores, they need to be adjusted to have the same range for each dataset. We use a scaling approach based on baseline methods, similar to that used by the Open Problems in Single-cell Analysis project[20]. We use four baseline methods: all features, 2,000 highly variable features selected using the batch-aware variant of the scanpy-Cell Ranger[21] method (as a representative commonly used approach suggested as good practice[3,22]), 500 randomly selected features (scores averaged over five feature

sets) and 200 stably expressed features selected using the scSEGIndex method[23] (as negative controls that should not capture signal) and use single-cell variational inference (scVI)[24] to integrate each dataset using the selected features. These methods are sufficiently diverse to demonstrate the effective range of each metric and allow us to establish baseline ranges for each dataset (Fig. 2a).

We scaled the metric scores using the baseline ranges and aggregated them as shown in Fig. 2, using the scIB pancreas dataset[3] as an example. This dataset was also used in stage 1 of the registered report. Along with the real baseline methods, we include theoretical 'Good' and

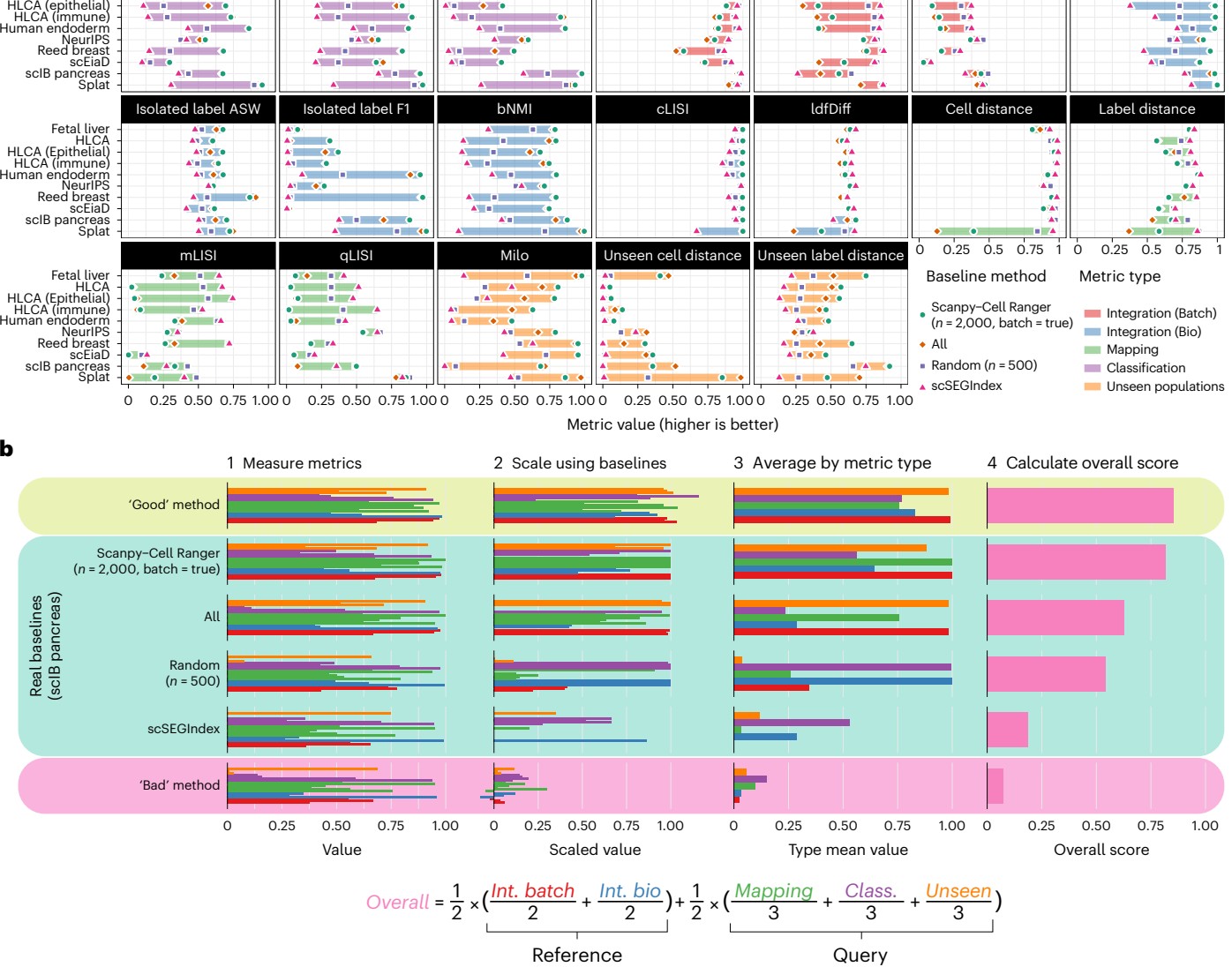

$$Overall = \frac{1}{2} \times \left( \frac{Int.\ batch}{2} + \frac{Int.\ bio}{2} \right) + \frac{1}{2} \times \left( \frac{Mapping}{3} + \frac{Class.}{3} + \frac{Unseen}{3} \right)$$

**Fig. 2 | Establishing baseline ranges and scaling and aggregating metrics.**
**a**, Baseline ranges for selected metrics. Each panel shows baseline scores for all datasets for a single metric. Shaded areas colored by metric type show the baseline ranges, and points show the values for individual baseline methods. **b**, The process for scaling and aggregating metrics using the scIB pancreas dataset as an example. The real baseline methods and theoretical 'Good' and

'Bad' methods are shown. First, the metrics are measured, and then the values are scaled using the baseline ranges. Scaled values greater than one or less than zero are possible if a method performs better or worse than the baselines. Average scores for each metric type are computed, and the overall score is calculated as a weighted average of the category scores using the equation below.

'Bad' methods that illustrate the behavior of methods that generally perform well or poorly across metric types (in contrast to the baselines, which each score highly on some metric types and lowly on others). The raw metric scores are scaled relative to the minimum and maximum baseline scores. After scaling, scores greater than one are possible if a method outperforms all the baselines (the 'Good' theoretical example) or negative scores are possible if a method performs worse than all the baselines (the 'Bad' theoretical example). The interpretability of scores outside the reference range is an advantage of this scaling approach, providing additional context to the scaled values. We calculated summary scores for each metric type by taking the mean of the scaled values for that category. A final overall score is calculated as a weighted mean of category scores (Fig. 2b).

We chose this weighting scheme to give equal importance to integrating the reference and mapping of the query and, within those,

equal consideration to the different metric types. While the overall scores are useful, we also present scores for each metric type in the following sections.

**The number of selected features affects performance**
In addition to the method used to select features, the number of selected features affects the success of integration and query mapping. Evaluating different feature set sizes for every selection method would be ideal but computationally prohibitive. Instead, we tested different numbers of features for a set of commonly used methods from the Seurat and scanpy packages, as well as simple methods that select the most expressed or variable features.

Figure 3a shows standardized summary scores (z-scores for each dataset and method combination), highlighting the trend with the number of features. We see different trends for categories that focus

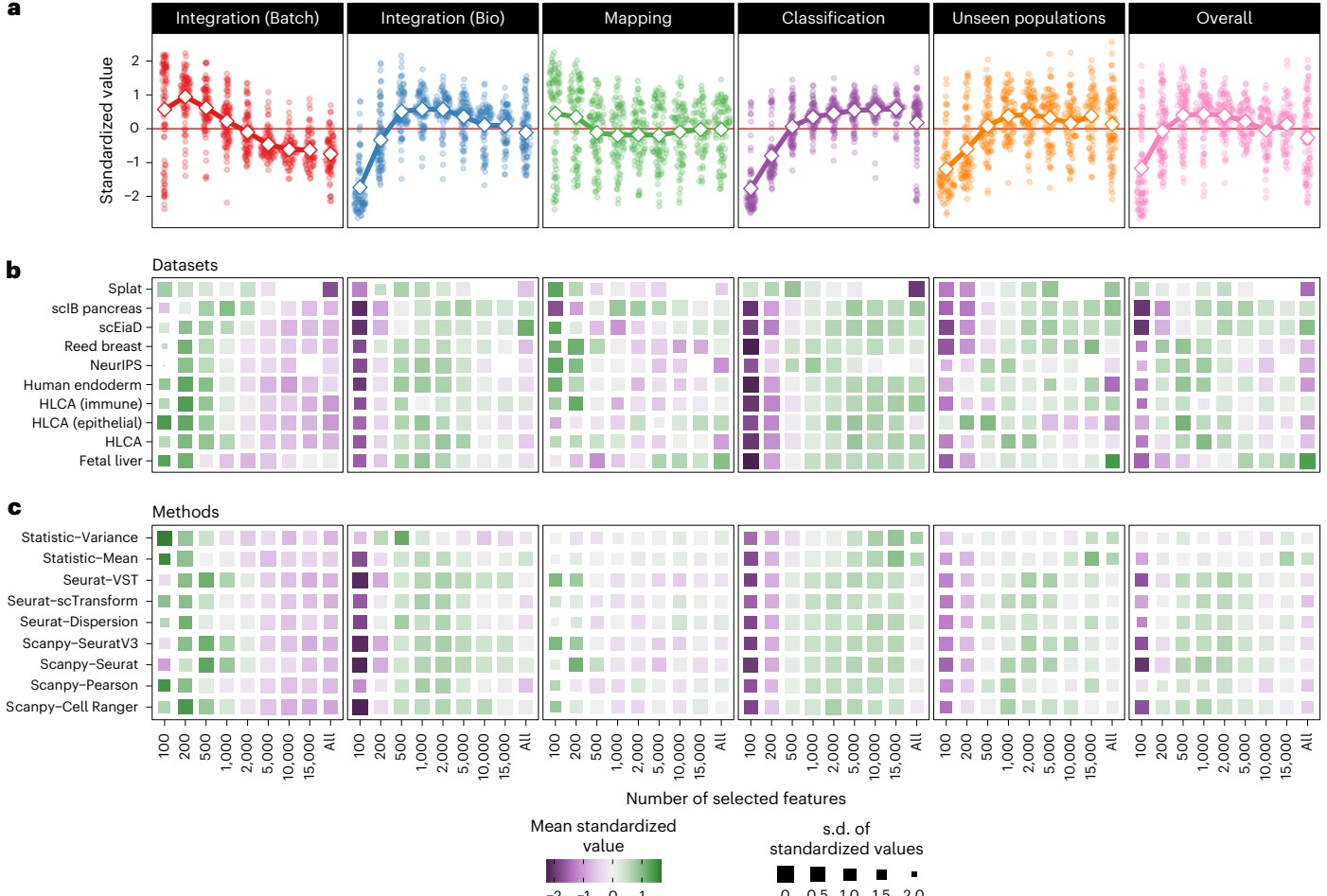

**Fig. 3 | Effect of the number of selected features on metric performance.**
**a**, Metric values standardized by dataset and method across different numbers of features for each metric category and overall scores. Points show individual standardized values and large diamonds connected by lines show the mean for each number of features. **b**, Heatmap of standardized values by metric type for each dataset (Extended Data Fig. 4a). Colors indicate mean standardized values and sizes of squares show the s.d. (smaller squares are more variable). Methods are ordered using hierarchical clustering. **c**, Similar heatmap to **b** but rows are methods rather than datasets (Extended Data Fig. 4b).

on batch correction than those that measure biological variation. The Integration (Batch) score shows the highest values for small feature sets and decreases as the number of features increases. The mapping category shows a similar but less extreme trend, converging to the mean value after around 500 features. The other categories show different patterns, increasing with the number of features before leveling off (classification and unseen populations) or declining (Integration (Bio)). These patterns reflect that achieving high scores for batch correction is possible by creating a noisy integrated embedding (a single noisy mass of cells). In this case, batches will be well mixed in the reference and the query, but there is no separation between cell types, resulting in low scores for the other categories. Due to this effect, we gave a lower consideration to the Integration (Batch) category when choosing the number of features. The overall score shows a similar trend to the biological categories, with peak values between 500 and 5,000 selected features.

While there are clear trends for each metric category, there is also significant variation. The following panels in Fig. 3 show mean standardized values for datasets band methods. We see that methods are largely consistent across datasets Fig. 3c. The Seurat-VST[25], scanpy-SeuratV3 and scanpy-Seurat methods peak at slightly higher numbers of features, whereas the statistic-Variance and statistic-Mean methods peak at lower numbers of features for Integration (Batch) and Integration (Bio) but higher numbers of features for classification and

unseen populations (Extended Data Fig. 4). This pattern suggests that selecting features in these simple ways can return sets that capture information well in the reference but not as well in the query compared to more sophisticated methods.

We see more variation in the highest-scoring number of features when methods are averaged for each dataset (Fig. 3b and Extended Data Fig. 4). The two datasets with the fewest cells (splat and scIB pancreas) show different patterns. For the simulated splat dataset[26], few features are required to capture the variation present. In contrast, the highest scores are associated with higher numbers of features for the scIB pancreas dataset. These differences reflect the properties of the two datasets, with the splat simulation producing data with less complexity than a real dataset, whereas the scIB pancreas dataset contains data from several technologies that present a difficult integration challenge. The larger fetal liver dataset also requires more features to achieve high scores in the query categories, with the highest averages for the mapping and unseen population categories when all features are used. This trend suggests that feature sets selected from the reference do not capture information in the query for this dataset. While less pronounced, this trend holds across all datasets, with more features required to achieve high scores on the classification and unseen population categories compared to the Integration (Bio) category; however, the performance of selecting all features shows a limit to how much additional signal can be obtained. The number of features

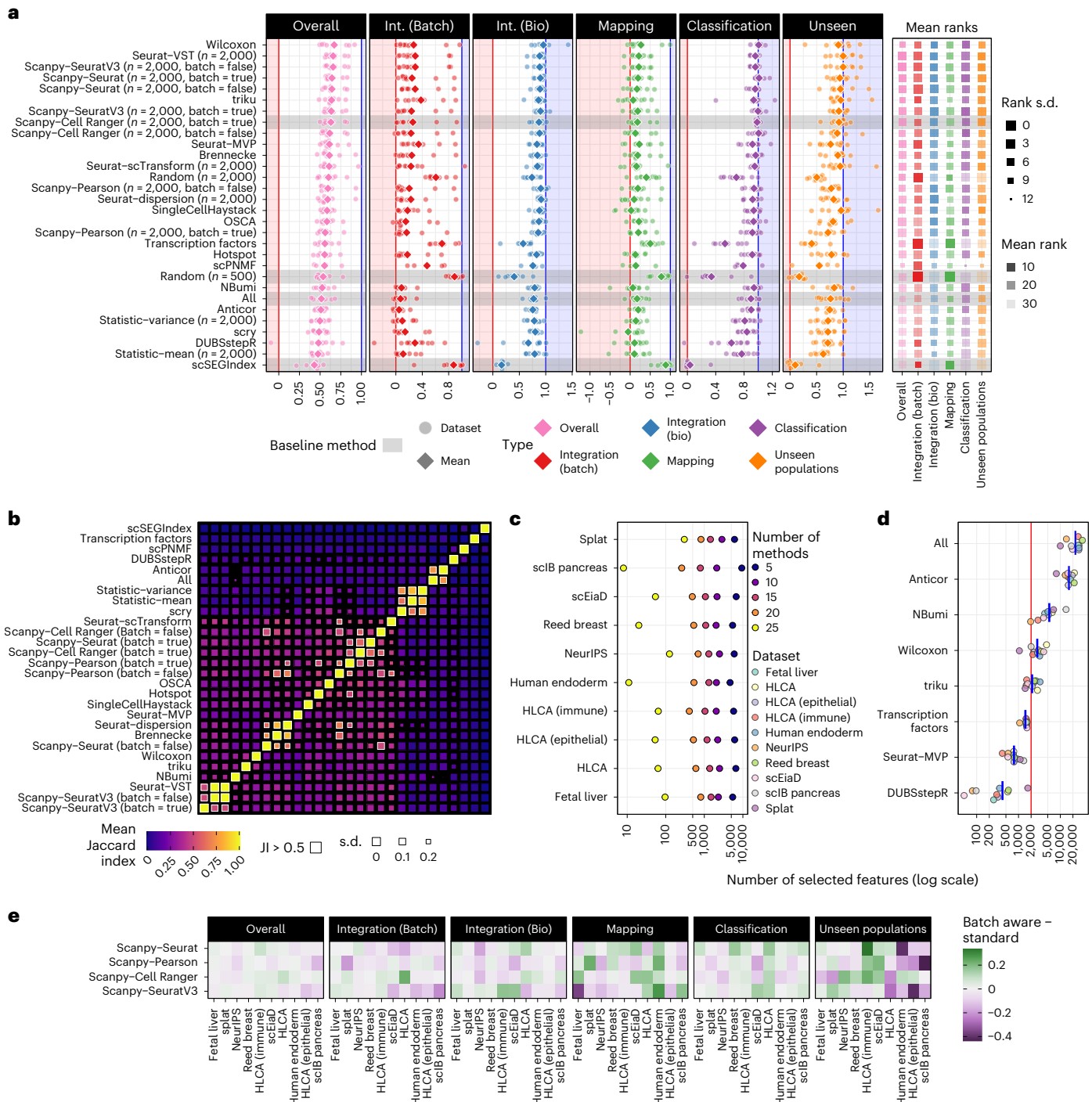

**Fig. 4 | Results of the benchmark of feature selection methods. a**, Summary of method performance by metric type. Points show scores for individual datasets and diamonds show the mean values (Extended Data Fig. 5a). Methods are sorted by mean overall score, and baseline methods are indicated by gray shading. Shaded areas show scores less than (red) or greater than (blue) the baseline range (0–1). Average rankings for each metric type are shown on the right, with color indicating mean rank and size s.d. (smaller is more variable) (Extended Data Fig. 5b). **b**, Overlap of features selected by different methods. The heatmap shows the mean Jaccard index (JI) between feature sets selected by different methods (excluding random gene sets) (Extended Data Fig. 6). Sizes of squares indicate the s.d. (smaller is more variable). Mean JI values greater than 0.5 are highlighted

with white borders. **c**, The number of features (on a $\log_{10}$ scale) selected by at least *n* methods (*n* = 25, 20, 15, 10 and 5) for each dataset. Colors indicate the number of methods. **d**, The number of features selected by different methods. Points are colored by dataset, and blue bars show the mean for each method. Only methods which automatically determine the number of features are shown. Most other methods were set to select 2,000 features, as indicated by the red line, except scPNMF, which uses 200 features. **e**, Heatmap of the relative performance of batch-aware variants of scanpy methods. Colors show the difference in score for each metric type on each dataset, with negative values (purple) indicating that the batch-aware variant performed worse than the standard approach and positive values (green) that it performed better.

at which the additional signal saturates is unclear and is likely to be different for each dataset as a function of the biological and technical diversity that is present.

Based on this analysis, we used 2,000 features for most methods in the following evaluation, as this number consistently produced high scores across datasets, methods and metric categories. Exceptions to this are methods that dynamically select the number of features (Anticor[27], DUBStepR[28], NBumi[29], Seurat-MVP[11] and triku[30]) and single-cell projective non-negative matrix factorization (scPNMF)[31], where the documentation recommends using fewer features than other methods for which we use 200 features.

### Highly variable features and supervised methods perform well
After determining the number of features to use, we compared feature selection methods. We were able to successfully run the majority of methods on all datasets; however, NBumi failed to complete on the Reed breast dataset[32] within 24 h, scPNMF, exceeded 400 GB of memory or failed to complete in 24 h on the Human Lung Cell Atlas (HLCA)[33], HLCA immune, HLCA epithelial, Human endoderm[34] and Reed breast datasets, and Anticor produced an unexpected error for the Human endoderm dataset.

Figure 4a shows the overall results for each metric category, sorted by the mean overall score across datasets for scVI integration (Extended Data Fig. 5a). Several methods obtain similar average overall scores. The Wilcoxon method, the only method to select features using cell labels, has the highest average overall score but is more variable across datasets than other top-performing methods. This higher variability suggests that supervised selection of features may not be effective for all datasets, even when the same labels are used for evaluation, and that tuning the number of features selected using this approach could be required. The Seurat-VST method obtains the highest overall ranking and several other highly variable feature selection methods also perform well with similar mean scores and more consistent performance than Wilcoxon. The other top-performing alternative method is triku, which has similar overall scores to the highly variable selection methods but shows some bias toward batch correction over conserving biological variation.

The lower-ranked methods show more variation in scores for individual categories (Extended Data Fig. 5). In particular, the baseline random and scSEGIndex methods score very highly on the Integration (Batch) and mapping categories but poorly on the categories measuring biological information. This effect demonstrates that it is easy to obtain good mixing between batches by selecting features that only contain noise and the importance of including metrics that measure the conservation of biological variation. Using a predefined list of transcription factors also produces a bias toward batch correction, demonstrating that it is not sufficient for features to be biologically important but that they must also be relevant to particular datasets. Transcription factors are typically lowly expressed and therefore noisy. Although the effect is less pronounced, some methods, such as OSCA[35] and singleCellHaystack[36], rank highly on Integration (Bio) but not on batch correction, with singleCellHaystack also scoring similarly to the top methods on unseen population detection. The singleCell-Haystack method uses Seurat-VST as a preprocessing step to create a principal-component analysis (PCA) space where the final features are selected but these additional steps do not lead to better performance than Seurat-VST alone.

We see some overlap in selected features for most methods, but there are very few combinations where the mean Jaccard index is above 0.5 (Fig. 4b and Extended Data Fig. 6). One pair that stands out is Seurat-VST and scanpy-SeuratV3, which produce identical sets. This overlap is unsurprising, given that they are different implementations of the same method, but it is reassuring to see consistency between packages using different programming languages. As the selected features are identical, any differences in performance we see between

these methods results from randomness in integration or metrics. The scanpy-Seurat and Seurat-MVP methods also implement the same approach but the scanpy implementation allows specifying the number of features, while the Seurat implementation selects the number of features dynamically using a threshold. There are also some differences in preprocessing steps, contributing to their lack of consistency.

Despite the lack of high overlap between selected feature sets, we still see a core set of features selected by most methods, with between 500 and 1,000 features being selected by at least 20 methods for most datasets (Fig. 4c). This consistency suggests that a subset of features clearly contains information for a dataset and should be crucial for effective integration and query mapping. That the remaining selected features are less likely to be shared between methods that have similar performance may result from redundancy in gene expression, with several genes carrying information about the same biological processes.

The number of features selected by dynamic methods (Fig. 4d) can also be related to performance. The Anticor method selects the majority of features in each dataset and, therefore, performs similarly to using all features. DUBStepR uses the most complex procedure of the methods compared here, resulting in very few selected features and low overall performance. However, DUBStepR scores relatively highly on biological metrics, suggesting that the features it selects are informative but insufficient to correct batch effects. The dynamic methods that perform well (Wilcoxon, triku and Seurat-MVP) select a number of features closer to the 2,000 features we chose to use for most methods. Seurat-MVP selects fewer than 2,000 features for all datasets and in comparison to scanpy-Seurat, which uses the same algorithm but is set to 2,000 features, Seurat-MVP has higher Integration (Batch) scores but similar Integration (Bio) performance. While fewer features are adequate for integrating the reference, the additional features included by scanpy-Seurat improve query classification and unseen population detection.

Feature selection can also be employed in a batch-aware fashion by selecting features for individual batches and combining the results, typically by choosing the features selected for the most batches. The intuition behind this approach is that it avoids selecting features that vary between batches but not between biological states within a batch. To assess the effectiveness of this approach, we included batch-aware variants of the scanpy methods. Figure 4e shows the difference in performance for each dataset and metric type compared to standard selection. We see significant differences in the summary scores for some scenarios, but this effect is inconsistent across either datasets or metric types, and the differences in the overall score are relatively small. For example, batch-aware selection improves the unseen population score for the HLCA (Immune) dataset but is significantly worse for the HLCA (Epithelial), Human endoderm and scIB pancreas datasets. The OSCA method also selects features in a batch-aware way but does not rank among the top-performing methods. While we do not rule out batch-aware feature selection as a useful approach, we cannot identify a scenario where it is consistently more effective than selecting features across batches.

### Lineage-specific feature selection and integration
An open question in large-scale integration projects is whether to integrate across the full diversity of cell states or to limit the complexity by subsetting to specific lineages or conditions. While we cannot fully address this question here, we can investigate some aspects by considering the three versions of the HLCA dataset.

Figure 5a shows the rankings for all methods for each HLCA subset, including the overall ranking and the ranking for each metric type. In general, these follow the trends we observed when considering all datasets, and we do not see any methods that consistently rank higher on the lineage subsets compared to the full dataset. To see whether the similar rankings across subsets resulted from selecting similar

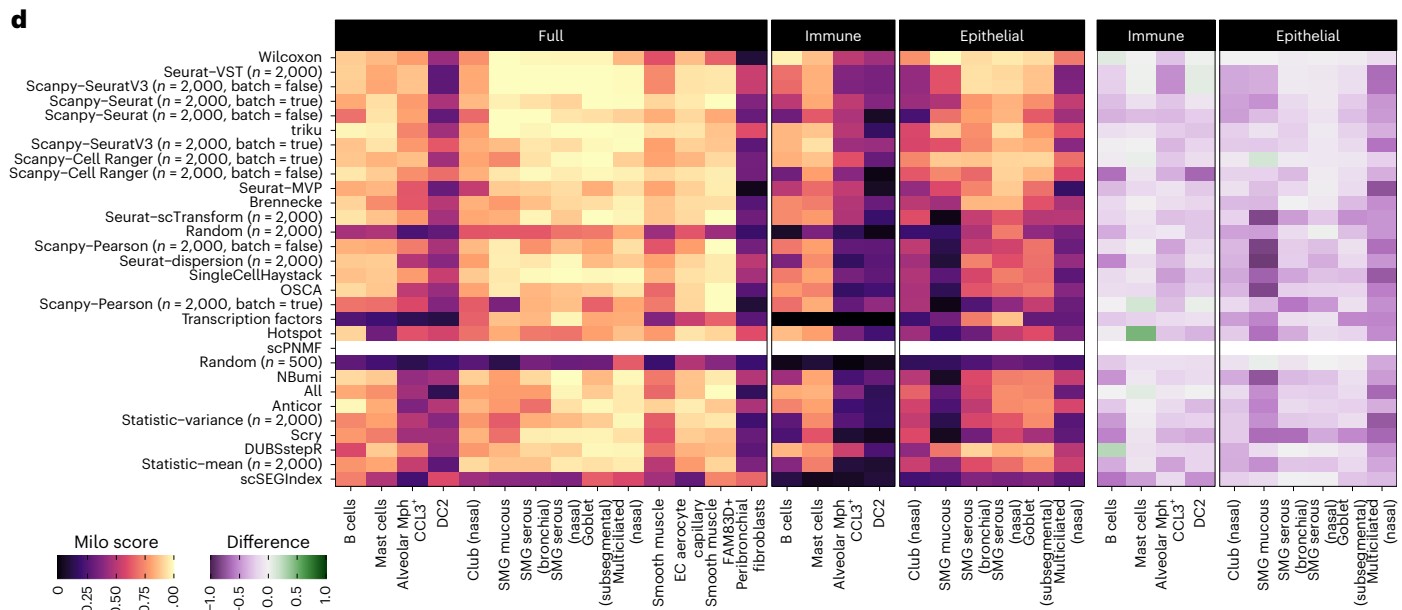

**Fig. 5 | Analysis of lineage subsets of the HLCA dataset. a**, Method rankings for the full HLCA dataset, the immune subset and the epithelial subset. Overall rankings are shown, along with rankings for each metric category. Methods are ordered by their overall performance across all datasets. **b**, Overlap of selected feature sets. The Jaccard index values between feature sets from each subset are shown as a heatmap. **c**, Overlap with marker genes. A heatmap of the mean proportion of marker genes selected by each method on each dataset subset.

The mean is calculated for each lineage in the full dataset (endothelial, epithelial, immune and stroma). The size of squares shows the s.d. of proportion across cell types in each lineage (smaller is more variable) (Extended Data Fig. 7). Overlaps are not shown for random gene sets. **d**, Analysis of cell label Milo scores. A heatmap shows the Milo score for each unseen cell type on the full, immune and epithelial subsets. On the right is shown the difference in scores for each lineage subset compared to the full dataset.

feature sets, we computed the Jaccard index between selected features (Fig. 5b). While there is some similarity in feature sets, the overlap is not higher than we saw between all datasets. The Jaccard index tends to be lower for higher-ranking methods, suggesting that these methods can successfully adapt to each dataset. We also see that the overlap in selected features between the immune and epithelial subsets is less than with the full dataset.

One motivation for lineage-specific feature selection is that it results in selecting more specific features for the cell types in that subset. To test this, we considered the published marker gene sets for the HLCA and calculated the proportion of these markers selected by each method on each dataset subset. Figure 5c and Extended Data Fig. 7 show the mean proportion of selected markers across cell types for each lineage in the full HLCA (endothelial, epithelial, immune and

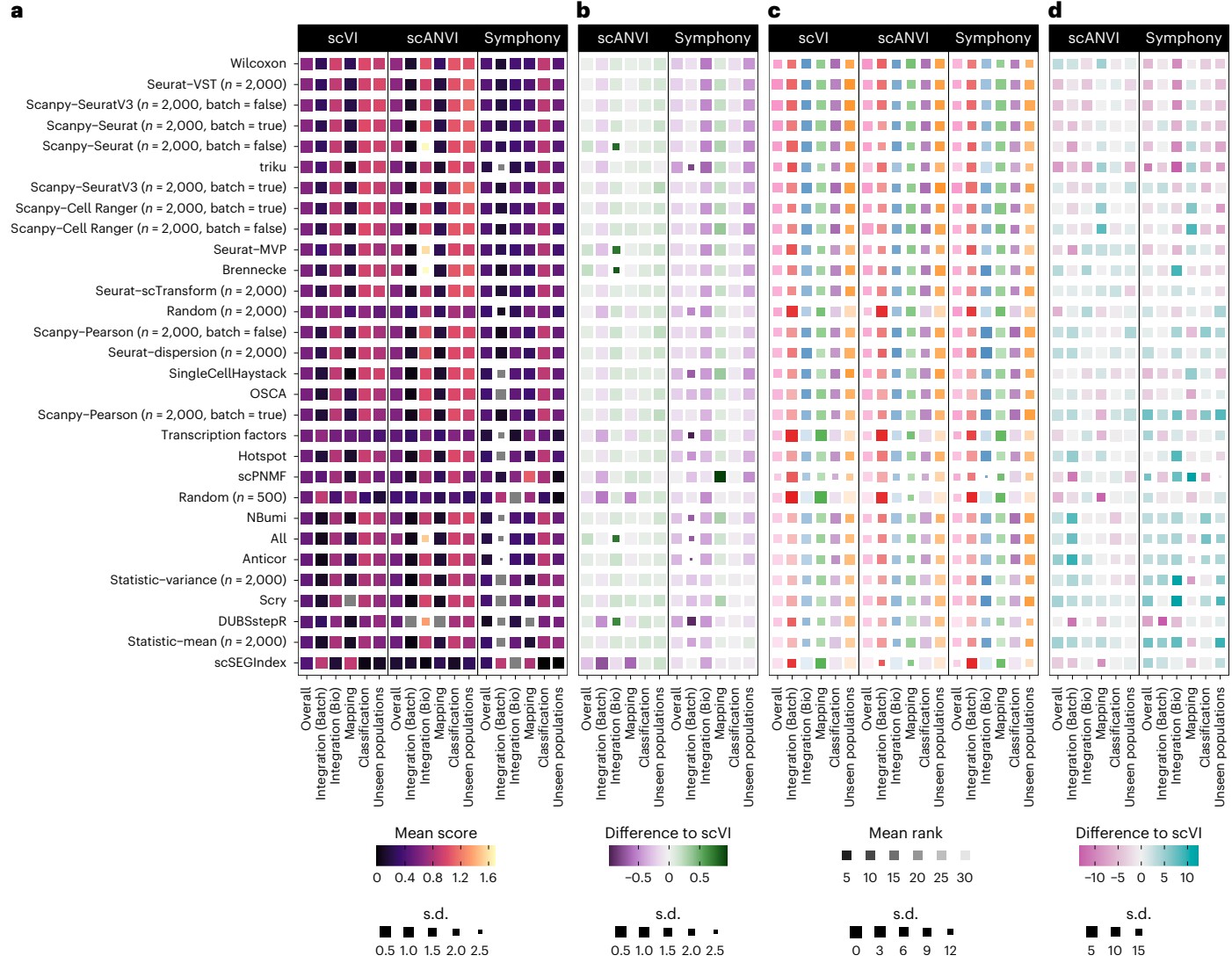

**Fig. 6 | Comparison of feature selection method performance for different integration and query mapping methods. a**, A heatmap of mean scores for each metric category for the evaluated methods for integration and query mapping with scVI, scANVI and Symphony (negative scores in gray). **b**, A heatmap of difference in mean scores for scANVI and Symphony compared to scVI. **c**, A heatmap of mean ranks for methods for each metric category. **d**, A heatmap of differences in mean ranks compared to scVI. In all heatmaps, colors represent values, and sizes of squares show s.d. across datasets (smaller is more variable). Methods are ordered by overall ranking for scVI.

stroma). In most cases, relatively few calculated markers are selected (proportion of markers mean, 0.38; median, 0.39; and first quartile, 0.04) (Extended Data Fig. 7). The lack of markers chosen may be due to redundancy in the information contained by related genes and differences in which features are prioritized for selection compared to marker detection. Selectivity of the markers chosen was not related to performance, with some of the worst-performing methods most effectively selecting markers only for the cell types in a specific lineage (Extended Data Fig. 7).

So far, we only considered the ranks of methods because individual scores are not directly comparable between subsets as they contain different cells and labels. To consider one area in more detail, we calculated Milo scores for individual unseen labels, allowing us to see if an unseen cell type is easier to distinguish in a whole-tissue or lineage-specific atlas (Fig. 5d). We see a clear trend of lower scores on the lineage subsets. This pattern supports the argument that by providing more diverse input data to the integration model it learns more of the possible cell space and can, therefore, better distinguish new cell populations.

**Interaction between selected features and integration method**

The focus of this study is the effect of feature selection rather than integration method, but we also measured the performance of the semi-supervised single-cell annotation using variational inference (scANVI) model[37] and Harmony[13] followed by query mapping using Symphony[12] (referred to as 'Symphony') in addition to scVI. This analysis allows us to assess the interaction between feature selection and integration models and the effect of biological supervision. Figure 6 shows the average scores and ranks for each integration method and the differences in performance for scANVI and Symphony compared to scVI.

Overall, there are no clear differences in metric rankings (Fig. 6d). We see a slight trend toward decreases in rankings for methods that rank highly for scVI and increases in rankings for methods that rank lowly for scVI (Extended Data Fig. 8). This effect could be explained by interactions between feature selection and integration methods or alternatively by scANVI and Symphony being less sensitive to feature selection or regression to the mean due to randomness in integration and some metrics. Looking more closely at the differences in scores (Fig. 6b), we see some methods that stand out. For scANVI,

there are significant improvements in the Integration (Bio) score for scanpy-Seurat (batch = false), Seurat-MVP, Brennecke[38], DUBStepR and all features. This improvement in performance showed that including biological information in the integration process can overcome the limitations of selected features in some cases.

In fact, scANVI leads to minor but consistent improvements for most metric types compared to scVI, except for Integration (Batch). This trade-off would be acceptable for many applications, particularly as the mapping score also increases, showing that preserving more biological information does not limit the ability to map query datasets to the reference. Symphony shows decreased performance compared to scVI across metric categories, except for the mapping score. While this decreased performance is relatively consistent across methods, the most significant decreases in the unseen population scores are for the highest-ranking methods. These results show that Symphony is unable to detect new cell populations that could be separated by scVI and scANVI using the same features.

## Discussion

In this comprehensive benchmark, we evaluated variants of 24 feature selection methods on ten datasets using 1,700 selected feature sets, over 6,000 integration runs producing over 140,000 metric scores. We performed a rigorous metric selection process and determined a number of features (2,000) that performed well across datasets. Our evaluation found highly variable feature selection methods to perform well, with the approach based on a variance-stabilizing transformation (Seurat-VST/scanpy-SeuratV3) being the top-ranked method. This result reinforces common practice and recommendations from previous benchmarks. Label-guided marker genes (Wilcoxon) also performed well but were more variable across datasets. We focused on unsupervised methods and other supervised techniques may produce more stable results; however, supervised feature selection only applies when cell labels are available, typically not the case before integration. The triku method was also highly ranked but showed some bias toward batch correction.

We did not find a consistent advantage for batch-aware variants of methods implemented in scanpy. Batch-aware selection could improve performance in some scenarios, but a more specific evaluation including additional methods is required to determine its applicability. For large datasets, batch-aware feature selection has a computational advantage, as loading the whole dataset into memory can be avoided. However, we could run many top-performing methods on the full datasets with relatively modest memory requirements.

We used scVI for our primary benchmark but compared the performance to scANVI, to inspect the effect of adding prior knowledge, and Symphony to see the interaction with an alternative integration approach. We saw that methods performed differently across integration approaches but did not identify clear relationships, suggesting that differences are the result of randomness in integration runs and shuffling between equally performing methods; however, there were clear differences between integration methods, with scANVI improving in all metric categories for the same feature sets. In contrast, Symphony showed decreased performance compared to scVI, particularly at unseen population detection.

Using subsets of the HLCA dataset, we considered lineage-specific feature selection. We did not see any clear preference for methods and particular lineages, and the top-performing methods effectively adapted to different subsets. Milo scores for individual unseen labels showed that it is easier to distinguish new cell populations using a more diverse reference atlas; however, this comparison was not our primary focus, and further work is required to determine if or when lineage-specific features selection and integration can be effective. For example, we did not consider whether lineage-specific features could improve integration of the full dataset or attempt to disentangle effects of feature selection from integration.

We only compared different numbers of features for some common methods to select a number of features for the final evaluation as the computation required was infeasible for all methods. For the methods where we examined different numbers of features, we observed a relationship between datasets and the optimal number of features for different metric types; however, the limited number of datasets did not allow us to connect this relationship to specific technical features, such as the number of batches or cell labels, and methods may perform differently with a different number of features. We encourage analysts to tune the number of selected features for their dataset and use case and we believe this will affect performance more than switching between top-performing methods; however, adjusting the number of features is computationally intensive and difficult to assess with new datasets as labels are typically not available for evaluation. Developing methods for automatically tuning the number of selected features based on technical aspects of datasets is a potential avenue for future research. We also emphasize that better performance on query tasks, especially unseen population detection, needs more features than producing a good integrated reference and should be considered if this is an intended use.

During the planning and implementation of this study, several feature selection methods[39–44], alternative metrics[45,46] and other comparisons[47–49] were published. While we consider it is unlikely that other methods would significantly improve performance, establishing this requires further benchmarking. More likely to affect the results is the inclusion of additional metrics, such as the recently proposed scGraph metric[46] which aims to address limitations of some metrics by considering distances between cell labels and has shown significant differences in performance between integration methods.

Our benchmark reinforces established practices as highly effective and provides guidance on generally effective parameters that can be optimized for individual datasets.

## Online content

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

## Methods

Our study follows a standard benchmark design, consisting of test datasets, feature selection methods to be evaluated and metrics for measuring performance (Extended Data Fig. 1). The complete benchmarking pipeline is implemented as a Nextflow[50] workflow (Extended Data Fig. 2) available from GitHub[51] and archived on Zenodo[52]. Summaries of the specific methods, metrics, datasets and processing steps are provided in the following sections. Please refer to the supplementary methods, pipeline code, original publications and package documentation for further information.

### Evaluated methods

We selected a range of feature selection methods covering approaches from standard analysis workflows and alternative methods proposed for scRNA-seq data. To be considered, a method must be implemented in a publicly available package that we could reliably install and run. Some methods can automatically determine the number of features to select, but for most others this must be specified. A few methods can consider batch labels during selection, but for most, this requires manually splitting the data, computing feature sets on each batch and combining the results. We have used the default settings or what is recommended in any accompanying documentation for most methods, but for a subset of highly used methods, we evaluated variants. Any preprocessing steps required before feature selection are considered part of the method. We used the steps suggested in the documentation for each method as they are recommended by the authors and represent the most likely real-world usage.

**Simple control methods.** We include all features and random feature sets in the evaluation as control methods. We expect that using feature sets selected by real methods improves performance over using all features and any randomly selected sets. To control for variability in selecting random features, we always include five random feature sets selected with different seeds and average metric scores over the five sets.

**Excess variability methods.** The most common approach to feature selection in RNA-seq analysis tool boxes such as scanpy[10] and Seurat[11] is to select highly variable features, those that show excess variability beyond what is expected. This approach assumes that extra variability results from differences in gene expression between cell populations or states and that selecting these features will identify those important to the cells in the sample.

We benchmark the following excess variability methods: features with the highest variance, the fitting method from Brennecke et al.[38] (implemented in scran[53] v.1.26.0), variants from Seurat[11] (v.4.3.0) (Seurat-dispersion, Seurat-MVP[11] and Seurat-VST[25]), variants from scanpy[10] (v.1.9.1) (scanpy-Seurat, scanpy-SeuratV3 and scanpy-Cell Ranger) and the approach from 'Orchestrating Single-Cell Analysis with Bioconductor'[35] using batchelor[54] (v.1.14.0) and scran. For scanpy methods we used both standard and batch-aware variants.

### Methods based on other statistical features

Other feature statistics can also be used for feature selection including selecting features with the highest mean expression, Anticor[27] (v.0.1.8), which selects features with excess negative correlations, NBumi which selects features with excess zeros (M3Drop v.1.24.0)[29] and DUBStepR (commit 76aa3948), which uses stepwise regression of a binned correlation matrix[28].

For Anticor, we disabled the filtering of predefined gene pathways as it requires gene identifiers, which are not available for all datasets. For NBumi, we select features with an adjusted $P$ value <0.01 unless this results in fewer than 500 features, in which case the 500 features with the lowest $P$ values were used.

### Model-based methods

Model-based methods fit an appropriate distributional model to the dataset. Features are then selected by looking for those significantly different from the fitted model. These include scTransform[55] (v.0.3.5, accessed via Seurat), analytic Pearson residuals[56] (implemented in scanpy) and scry (v.1.10.0)[57].

### Embedding-based methods

Dimensionality reduction is a commonly used preprocessing step in scRNA-seq analysis. Some feature selection methods either use sophisticated embedding methods or look for features that vary across an embedding. scPNMF (commit 47d5b10c) performs a modified PNMF, where an alternative initialization is used and selects features associated with informative bases[31], and singleCellHaystack (v.0.3.4) uses Kullback-Leibler divergence to find features that are expressed in subsets of nonrandomly positioned cells[36]. For singleCellHaystack, we first select features using Seurat-VST and perform a 50-dimensional PCA as input.

### Graph-based methods

Another common step in scRNA-seq analysis is to build a nearest-neighbor graph of cells, typically using positions in an embedded space. Some methods operate on these graphs. Hotspot (v.1.0.0) looks for features with a high local auto-correlation within a graph[58] and triku (v.2.1.4) uses a neighborhood graph to distinguish features that are expressed in a few cells randomly across a dataset from those that are expressed in a few related cells[30]. For both, we use a graph based on a PCA of all features as input.

### Supervised methods

We focus on evaluating unsupervised feature selection methods, as cell labels are typically not available before the integration process; however, at least some level of cell labels may be available, particularly for atlas-building projects that combine previously annotated public datasets. As an example supervised method, we include marker genes selected using the Wilcoxon rank-sum test (as implemented in scanpy) followed by a filtering procedure to remove features expressed in less than 10% of cells within a label, expressed in more than 80% of cells outside the label or with a $P$ value >0.1. The remaining features are sorted by estimated log fold change and the top 200 features are selected per label. The final feature set is the intersection of the features selected for each label.

We also included known transcription factors downloaded from The Human Transcription Factors[59] website (https://humantfs.ccbr. utoronto.ca/index.php) selecting 1,639 genes where the 'Is TF?' field was equal to 'Yes'. The intersection of this list with the genes in each dataset was used. This method cannot be applied to the splat dataset as it does not contain real gene names.

### Stable expression methods

The opposite of highly variable features are those stably expressed or varying less than expected. The scSEGIndex method in the scMerge package (v.1.1.4.0) fits a gamma-Gaussian mixture model to each feature[23]. The parameters of this model and other features, such as the proportion of zero counts, are used to rank features and calculate a stability index. We used these stable features as a negative control and they should perform poorly for integration as they should not capture either technical noise or biological signal.

### Evaluation metrics

We implemented a wide array of metrics designed to evaluate different aspects of creating and using an integrated scRNA-seq reference. Some metrics require a ground truth cell label, while others are unsupervised and measure whether the structure in a single sample is maintained. All metrics are designed so that a raw score of 0 represents the worst possible performance and a raw score of 1 the best possible performance.

**Integration (Batch).** Integration (Batch) metrics measure the mixing between batches in the reference. Cells of the same cell type should be thoroughly mixed and neighborhoods should be equally likely to contain cells from any batch. The batch ASW[3], Batch PCR[3], graph connectivity[3] and graph-based iLISI[3,13] are implemented in scIB[3] (v.1.1.4) using scikit-learn[60] (v.1.1.2). The kBET metric[17] is accessed from the kBET R package (commit a10ffeaa) via scIB. To calculate an overall score for the Seurat mixing metric[14] we divided the cell scores by the maximum neighborhood size, took the mean across cells and subtracted from 1 so higher scores are better. For the CMS metric[18] in the CellMixS package (v.1.14.0) we use 1 minus the proportion of cells with a $P$ value <0.1.

**Integration (Bio).** Integration (Bio) metrics measure whether biological signals (primarily cell labels) are conserved after integration. Unlike batch correction metrics, where perfect scores can be obtained by mapping cells to a single point, biological conservation metrics require that cell labels are separated after integration. The label ASW[3], graph-based cLISI[3,13], cell cycle conservation[3], ARI[3], NMI[3], Isolated labels ASW[3] and Isolated labels F1[3] metrics are implemented in scIB using scikit-learn. bARI[16] and bNMI metrics are available from balanced_clustering (commit a2ae3a4d). For the Seurat local structure metric[14] we used the average over all cells as the final score and for ldfDiff[18] we took the absolute distance and set an upper bound to get a cell score and used 1 minus the mean cell score as the overall score. The cell cycle metric[3] scores cells[11] using genes from Tirosh et al.[61] with ENSEMBL IDs obtained from Biomart[62] using the biomaRt package[63]. It cannot be calculated for the splat dataset as it does not contain cell cycle effects. For metrics that require clusters (ARI, NMI, bARI and bNMI), we performed Leiden clustering with the resolution parameter set to values between 0.1 and 2 in steps of 0.1 using scanpy via scIB and selected the resolution with the best metric score.

**Mapping quality.** Mapping quality metrics assesses how well the reference represents the query and is able to merge it into the same space. For perfect mapping, cell types present in both the reference and query should be mixed, as should batches within the query. At the same time, biology within the query should be preserved. The cell distance metric calculates the Mahalonobis distance between each mapped query cell and the distribution of the corresponding label in the reference[12]. To create a bound for the distance we calculate the distance for every cell in the reference for a label and take the 90th quantile. The final score is 1 minus the proportion of mapped cells outside the boundary. The label distance considers labels as a whole rather than individual cells[12]. The Mahalonobis distance is calculated between the centroid of the label in the query and the matching label in the reference. Labels are skipped if they have fewer than 20 cells in the query or are not in the reference. We used the maximum distance of query cells to their label centroid as a boundary. Distances to the matching reference label are then scaled using this value and set to 1 if they exceed the maximum distance. The final score is the mean across cell types.

mLISI is the same as iLISI but measures mixing between the query and reference (also known as ref_query LISI[12]) and qLISI measures mixing between query batches after mapping (also known as query_donors LISI[12]).

$k$NN correlation measures how well cell neighborhoods are maintained[12]. For each query batch, a PCA is performed and the Euclidean distances to the 100 nearest neighbors of each cell are calculated. The distances to the same neighbors in the joint integrated embedding are also calculated and the Spearman correlation is computed. After adjusting the correlations to the range 0 to 1, the mean of cells in each batch is calculated and the final score is the mean across batches. For particularly bad integrations (that is small random feature sets), a cell may be equally distant from all neighbors, in which case the correlation cannot be calculated and it is assigned a score of 0.

The reconstruction metric assesses a generative model's ability to represent query cells by sampling from the posterior distribution and measuring the cosine distance between the mean posterior expression profile and the true cell expression profile[64]. We adjusted the distances to be in the range 0 to 1 and took 1 minus the mean distance as the final score. This metric cannot be calculated for Symphony integrations as it is not a generative method.

**Classification.** The classification (or label transfer) metrics measure how well a classifier trained on the reference can correctly predict labels for query cells. We use standard classification metrics: accuracy, F1 score, Jaccard index, Matthews correlation coefficient (adjusted to [0, 1]) and macro-averaged area under the precision-recall curve as implemented by scikit-learn. For F1 and the Jaccard index we use micro, macro and rarity-weighted[19] averages over labels.

**Unseen population prediction.** Unseen population metrics focus on novel biology in the query by measuring how mapping has affected cell labels present in the query but deliberately left out of the reference. These should be maintained as separate populations but an integration that does not properly capture variation may merge them with other labels.

The unseen uncertainty metric uses the output of the label transfer classifier and measures poor classification of unseen cell by calculating 1 minus the mean probability of the assigned class for query cells from unseen populations. Unseen cell distance is based on the cell distance metric but calculated only for unseen query populations. As the label does not exist in the reference, we calculate distances to each cell's nearest reference population and subtract the final score from 1 so that higher distances (greater separation from the reference) give higher scores. Unseen label distance applies similar changes to the label distance metric by calculating distances to the nearest reference label centroid.

We use the milopy[65] (commit be1a6cc8) implementation of the Milo differential abundance method[15] as a metric to detect unseen populations by taking query or reference as the covariate of interest[64]. A neighborhood graph is calculated in the integrated embedding using a number of neighbors equal to five times the number of batches (up to a maximum of 200). Milo is then applied to a subset of cells (up to 20,000 cells or 10% of the datasets, whichever is higher). The score for each label is the proportion of cell neighborhoods significantly associated with the query (false discovery rate-adjusted $P$ value <0.1). The overall score is the average of the proportions across all unseen labels. In rare cases for poor integrations where Milo cannot select cells from an unseen label, that label is assigned a score of 0.

### Benchmarking datasets

We selected datasets representing different scenarios (tissues, technologies and developmental stages) where integration is a critical analysis step, including smaller-scale datasets and larger atlases. We chose query batches by selecting batches with shared characteristics different from the remaining samples, such as technology, time point or location. The unseen populations removed from the reference were chosen by looking for labels enriched in the query batches and selecting labels presenting different challenges, such as rare or perturbed cells. For each dataset, we use the cell labels assigned by the original authors.

**scIB Pancreas.** We downloaded the scIB pancreas dataset[3] from figshare[66]. Cell labels were taken from the 'celltype' cell annotation column (12 reference labels) and batches from the 'tech' column. For the query, we used batches representing the CEL-seq and CEL-seq2 technologies with the 'activated_stellate' label treated as an unseen population. The prepared dataset contained 18,319 features, 12,731 reference cells (seven batches) and 3,243 query cells (two batches).

**NeurIPS 2021.** We downloaded the NeurIPS 2021 CITE-seq dataset[67,68] from the Gene Expression Omnibus (GEO)[69] (GSE194122) and used only the gene expression features. Cell labels were taken from the 'cell_type' annotation and batch labels from the 'batch' annotation. We considered samples from Site 4 as the query with the 'CD8+ T naive' and 'Proerythroblast' labels treated as unseen query populations. After preparation, the dataset contained 13,953 features, 70,061 reference cells (nine batches) with 42 reference labels and 16,715 query cells (three batches).

**Fetal liver hematopoiesis.** We downloaded the fetal liver hematopoiesis[70] dataset from CellAtlas.io[71] using batch labels from the 'fetal.ids' annotation and cell labels from the 'cell.ids' annotation. Three samples from different developmental stages were treated as the query with 'Kupffer Cell', 'NK', 'ILC precursor' and 'Early lymphoid_T lymphocyte' as unseen populations. The prepared dataset contains 26,686 features, 62,384 reference cells (11 batches and 23 reference labels) and 26,449 query cells (three batches).

**Reed breast.** We downloaded the version of the Reed breast dataset[32] released with the preprint[72] from the Chan Zuckerberg CELLxGENE: Discover Census (https://cellxgene.cziscience.com/)[73] (dataset ID 0ba636a1-4754-4786-a8be-7ab3cf760fd6, Census version 2023-07-05) using the cellxgene-census package (v.1.0.1) and subsetted to cells with a BRCA status of either wild-type ('WT' or 'assumed_WT') or 'BRCA1'. Donor ID was used as the batch label, with cell labels taken from the 'level2' annotation. We excluded a subset of cells labeled as doublets, as it is not clear how they should be treated by metrics. Wild-type cells were used to create the reference and BRCA1 cells were used as the query. The 'BSL2', 'CD8T 1', 'CD8T 2', 'CD8T 3', 'FB5', 'LEC1' and 'LEC2' labels were used as unseen labels. After preparation, the dataset contained 33,691 features, 337,339 reference cells (24 batches and 32 reference labels) and 197,649 query cells (17 batches).

**Single-cell Eye in a Disk.** We downloaded the single-cell Eye in a Disk (scEiaD) dataset[74] from the plae: PLatform for Analysis of scEiad website (https://plae.nei.nih.gov/) and selected the human cells derived from tissue samples where the organ was specified as 'Eye'. We removed cells that did not have a cell label or were labeled as doublets and batches with fewer than 500 cells remaining, as these caused some metrics to produce unreliable results. Cell labels were taken from the 'CellType_predict' annotation (harmonized labels from a classifier) and the 'batch' annotation was used for batches. We split batches using cell capture technology, with 10x v.2 taken as the reference and 10x v.3 and Drop-seq batches as the query. The 'B-Cell', 'Blood Vessel', 'Macrophage', 'Pericyte', 'Smooth Muscle Cell' and 'T/NK-Cell' labels are unseen populations. After preparation, the dataset contained 19,560 features, 360,270 reference cells (69 batches and 41 reference labels) and 48,496 query cells (18 batches).

**Human endoderm.** We downloaded the Human endoderm dataset[34] from Mendeley Data[75]. Individuals were treated as batches with labels obtained from the 'Cell_type' annotation. A small number of cells labeled as 'Undefined' were removed. Samples from weeks 12–15 were selected as the query with 'Basal like', 'Ciliated', 'Hepatocyte', 'Mesenchyme subtype 4' and 'T cell/NK cell 1' labels treated as query-specific. The prepared dataset consisted of 27,855 features, 100,580 reference cells (ten batches and 21 reference labels) and 44,784 query cells (four batches).

**Human Lung Cell Atlas.** We downloaded the core Human Lung Cell Atlas dataset[33] from the Chan Zuckerberg CELLxGENE: Discover Census (dataset ID 066943a2-fdac-4b29-b348-40cede398e4e, Census version 2023-07-25) and used the 'dataset' annotation as defined by the authors as batch labels with the 'ann_finest_level' annotation as labels. Datasets from organ donors were treated as the reference and healthy and diseased samples from living donors made up the query.

'Multiciliated (nasal)', 'Club (nasal)', 'Goblet (subsegmental)', 'SMG serous (nasal)', 'SMG serous (bronchial)', 'SMG mucous', 'EC aerocyte capillary', 'Peribronchial fibroblasts', 'Smooth muscle', 'Smooth muscle FAM83D+', 'B cells', 'DC2', 'Alveolar Mph CCL3+' and 'Mast cells' labels are unseen populations. After preparation, the dataset included 27,987 features, 314,573 reference cells (nine batches and 47 reference labels) and 251,400 query cells (five batches).

**HLCA (immune).** The HLCA (immune) dataset takes the full HLCA dataset and uses the coarsest level of annotation to select cells in the immune compartment. The batches and labels are the same as the full HLCA dataset, but after subsetting, only 'B cells', 'DC2', 'Alveolar Mph CCL3+' and 'Mast cells' remain as unseen labels. We also removed some batches with insufficient cells. The prepared dataset has 26,618 features, 155,385 reference cells (seven batches and 16 reference labels) and 52,795 query cells (two batches).

**HLCA (epithelial).** The HLCA (epithelial) dataset is a second subset of the HLCA dataset focusing on the epithelial compartment. This subset consists of 27,673 features, 118,374 reference cells (eight batches and 17 reference labels) and 162,875 query cells (five batches) with 'Multiciliated (nasal)', 'Club (nasal)', 'Goblet (subsegmental)', 'SMG serous (nasal)', 'SMG serous (bronchial)' and 'SMG mucous' remaining as unseen labels.

**splat.** Simulations address some limitations of real data by providing a definite ground truth. We generated a dataset using a modified version of the splat simulation in the Splatter package[26] designed to represent a scenario where a tissue is measured using three different technologies (two batches each) in two conditions. These 'technologies' measure a medium number of cells at medium depth (Batch1 and Batch2), a low number of cells at high depth (Batch3 and Batch4) and a high number of cells at low depth (Batch5 and Batch6), with the low-depth samples used as the query. The simulation contains ten cell labels, including a progenitor differentiating along two trajectories (one with an 'Intermediate' cell type only present in the query) and six discrete cell types that differ in number of cells, number of differentially expressed genes and number of detected features. The discrete groups include a 'Rare' population and a 'Perturbed' state, which are only present in the query. To increase the variability in the simulation, we added additional label-specific noise factors to the model, which were applied before generating counts. The splat dataset contains 9,984 features, 30,041 reference cells (four batches and seven reference labels) and 69,936 query cells (two batches).

### Benchmarking pipeline

To improve reproducibility, make sure that results are up-to-date as code is updated and easily take advantage of computing resources, we built a pipeline using Nextflow[50] (Extended Data Fig. 2). The pipeline takes a dataset, applies standard preprocessing and splits it into reference and query samples. The feature selection methods are applied to the reference, and selected features used for integration. After integration, the query is mapped to the reference, and a cell label classifier is trained. The reference and query, ground truth cell labels and transferred labels are provided to metrics. The metric scores are then scaled, aggregated and ranked. Pipeline stages use both Python (v.3.9.13) and R[76] (v.4.2.2), including packages from Bioconductor[77]. The Python anndata package[78] (v.0.8.0) was used to store data and save it as H5AD files between pipeline stages. The zellkonverter package (v.1.8.0) was used to load data into R via the reticulate (v.1.26) interface where it was stored as SingleCellExperiment[35] (v.1.20.0) or SeuratObject (v.4.1.3) objects.

### Dataset preprocessing

The preprocessing step includes basic quality control filtering of cells using scanpy and storing information (such as batch and label) in

standard locations. We removed cells with fewer than 100 total counts or expressing fewer than 100 features. The dataset is split into a reference and query based on the batch labels. Labels with fewer than 20 cells are removed from both the reference and query, as some metrics can behave unpredictably with small cell numbers. Labels defined as unseen populations are also removed from the reference. The final preprocessing step removes any features not expressed in the reference.

### Integration and query mapping

The base model we use for integration is scVI[24] available in scvi-tools[79] (v.0.17.1). This model uses a conditional variational autoencoder and allows the mapping of query samples using architecture surgery[80]. We also train a scANVI model[37] a semi-supervised extension of scVI where cell labels are used to finetune the network. These models take raw count data as input, so we did not consider the interaction between feature selection and normalization methods.

As an alternative approach based on correcting a PCA space, we included integration with Harmony[13] followed by query mapping using Symphony[12]. We provide Harmony with normalized expression values rather than raw counts as suggested by the documentation. Counts are first normalized to counts per 10,000, then log-transformed. The dataset is subset to the selected features and scaled with a maximum value of 10 (per feature) and 30 principal components are provided to Harmony. For Symphony, log-transformed normalized query data are provided (scaling is performed during mapping). Data preprocessing steps are performed using functions in scanpy and integration and query mapping are performed using harmonypy[81] (v.0.0.9) and symphonypy[82] (v.0.2.1).

### Label transfer

We trained a multinomial logistic regression classifier on the integrated reference using scikit-learn, taking the position of each cell in the integrated embedding space as input and the ground truth cell labels as the output. Labels are transferred to the query by providing the mapped embedding coordinates to the trained classifier, predicting the probability for each reference label and recording the label with the highest probability.

### Metric selection

For metric selection we used different numbers of randomly selected features across all test datasets. We also included feature sets of different sizes from the scanpy-Seurat method to evaluate the relationship with the number of features as random gene sets have no inherent ordering (the first features selected are no more informative than the last features selected). We evaluated the behavior of individual metric scores and the relationships between them. Metrics were removed if they could not distinguish between feature sets (have an insufficient dynamic range), were overly correlated (Pearson correlation) with the number of features, were associated with technical dataset features or showed undesirable correlation patterns.

### Selecting a number of features

We evaluated different numbers of features for methods in Seurat and scanpy as well as high variance or high mean expression. We calculated $z$-scores across methods and datasets to see how performance changed with the number of features. To reduce the computational cost, we limited this part of the analysis by methods rather than datasets as it allowed us to see the effect of the number of features across datasets. The number of features used for the benchmark (2,000) was chosen by considering trends over methods, datasets and metric types.

### Analysis of results

The relative rather than absolute performance of methods and the aggregation across metrics are most informative. All metrics produced scores in the range of 0 to 1 (with higher being better), but they have different real dynamic ranges. To scale each metric for each dataset we used a set of reference methods to establish the effective range of each metric. These are all features, randomly selected features, stably expressed features from scSEGIndex and batch-aware features from scanpy-Cell Ranger as an example of current standard practice[3,22]. Depending on the metric, using all features performs either well or poorly, while random and stably expressed features result in high batch-correction scores but poor biological conservation. The baseline methods were used to establish a range for each metric (for a dataset), and then all scores were scaled relative to that range. Scaling using baseline methods provides ranges that are more interpretable and are not affected by adding or removing methods.

The scaled metric scores were aggregated by taking the mean for each category. This level of aggregation gives a summarized performance for each of the methods for each task. An overall score for each dataset is obtained using a weighted mean of the task scores.

$$\text{Overall} = \frac{1}{2} \times \left( \frac{\text{Int.Batch}}{2} + \frac{\text{Int.Bio}}{2} \right) + \frac{1}{2}$$
$$\times \left( \frac{\text{Mapping}}{3} + \frac{\text{Class.}}{3} + \frac{\text{Unseen}}{3} \right)$$

Methods were ranked at the level of metric categories, datasets and over the whole benchmark. These rankings let us evaluate which methods perform better at different tasks or scenarios. We also checked for consistency between integration approaches and variants of feature selection methods.

Further analysis examined the similarity between methods by considering the overlap in selected sets calculated using the Jaccard index. We also compared between the full HLCA dataset and subsets representing the immune and epithelial compartments.

Final figures were produced using the ggplot2 package[83] (v.3.5.0) and assembled using patchwork (v.1.2.0). Data processing was performed using tidyverse[84] (v.2.0.0) packages.

### Reporting summary

Further information on research design is available in the Nature Portfolio Reporting Summary linked to this article.

## Data availability

All real scRNA-seq datasets were downloaded from public repositories provided by the original authors as described in the methods (scIB Pancreas, figshare[66]; NeurIPS, GEO (GSE194122); Fetal liver, CellAtlas.io[71]; Reed Breast, Chan Zuckerberg CELLxGENE: Discover Census (dataset ID 0ba636a1-4754-4786-a8be-7ab3cf760fd6, Census version 2023-07-25); scEiaD, plae: PLatform for Analysis of scEiad website (https://plae.nei.nih.gov/); Human endoderm, Mendelay Data[75]; and HLCA, Chan Zuckerberg CELLxGENE: Discover Census (dataset ID 066943a2-fdac-4b29-b348-40cede398e4e, Census version 2023-07-25)). Raw and prepared dataset files, selected feature sets, metric scores and rendered analysis reports from this benchmark are available from figshare[85].

## Code availability

All code associated with this study is available on GitHub[51] and archived on Zenodo[52], including scripts for downloading datasets from public repositories provided by the original authors, running methods and calculating metrics, the Nextflow pipeline and associated environment and configuration files. The code for analyzing the benchmark results, including the production of final figures, is also available in this repository.

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

## Acknowledgements

We acknowledge support from the Bavarian Ministry of Science and the Arts in the framework of the Bavarian Research Association 'ForInter' (Interaction of human brain cells) (F.J.T.) and the Chan Zuckerberg Initiative via grant CZIF2022-007488 - Human Cell Atlas Data Ecosystem (M.D.L. and F.J.T.). S.R. is supported by the Helmholtz Association under the joint research school Munich School for Data Science. O.D. received financial support from the Helmholtz Information and Data Science Academy to enable a short-term research stay at Helmholtz Munich. We thank J. Gagneur (Technical University of Munich, Helmholtz Munich, Munich Center for Machine Learning) and his administrative team, particularly F. Hölzlwimmer, for their generous support in providing access to computational resources. We also thank L. Heumos, S. Jimenez, Y. Ji, K. Hrovatin and F. Curion for their comments on drafts of the manuscript.

## Author contributions

L.Z. conceptualized the study, developed methodology used in the study, developed code for the study, performed the benchmark experiments, performed formal analysis of results, visualized results and created figures, wrote the first draft of the manuscript and edited and reviewed the manuscript. S.R., W.W., O.D. and A.F. contributed code and reviewed and edited the manuscript. R.K.-R. performed formal analysis of results and reviewed and edited the manuscript. L.V. performed benchmark experiments and reviewed and edited the manuscript. M.L. contributed to the design of the study and reviewed and edited the manuscript. F.J.T. supervised the study, acquired funding to support the study and reviewed and edited the manuscript. All authors read and agreed to the final manuscript.

## Funding

## Competing interests

F.J.T. consults for Immunai, Singularity Bio, CytoReason and Cellarity and has an ownership interest in Dermagnostix and Cellarity. A.F. is currently an employee of CytoReason. L.Z. has consulted for Lamin Labs, was an employee of iOmx Therapeutics and is currently an employee of Data Intuitive. R.K.-R. has consulted for iuvando Health. M.D.L. consults for CatalYm, has contracted for the Chan Zuckerberg Initiative and has received speaker fees from Pfizer and Janssen Pharmaceuticals. The other authors declare no competing interests.

## Additional information

**Extended data** is available for this paper at https://doi.org/10.1038/s41592-025-02624-3.

**Correspondence and requests for materials** should be addressed to Fabian J. Theis.

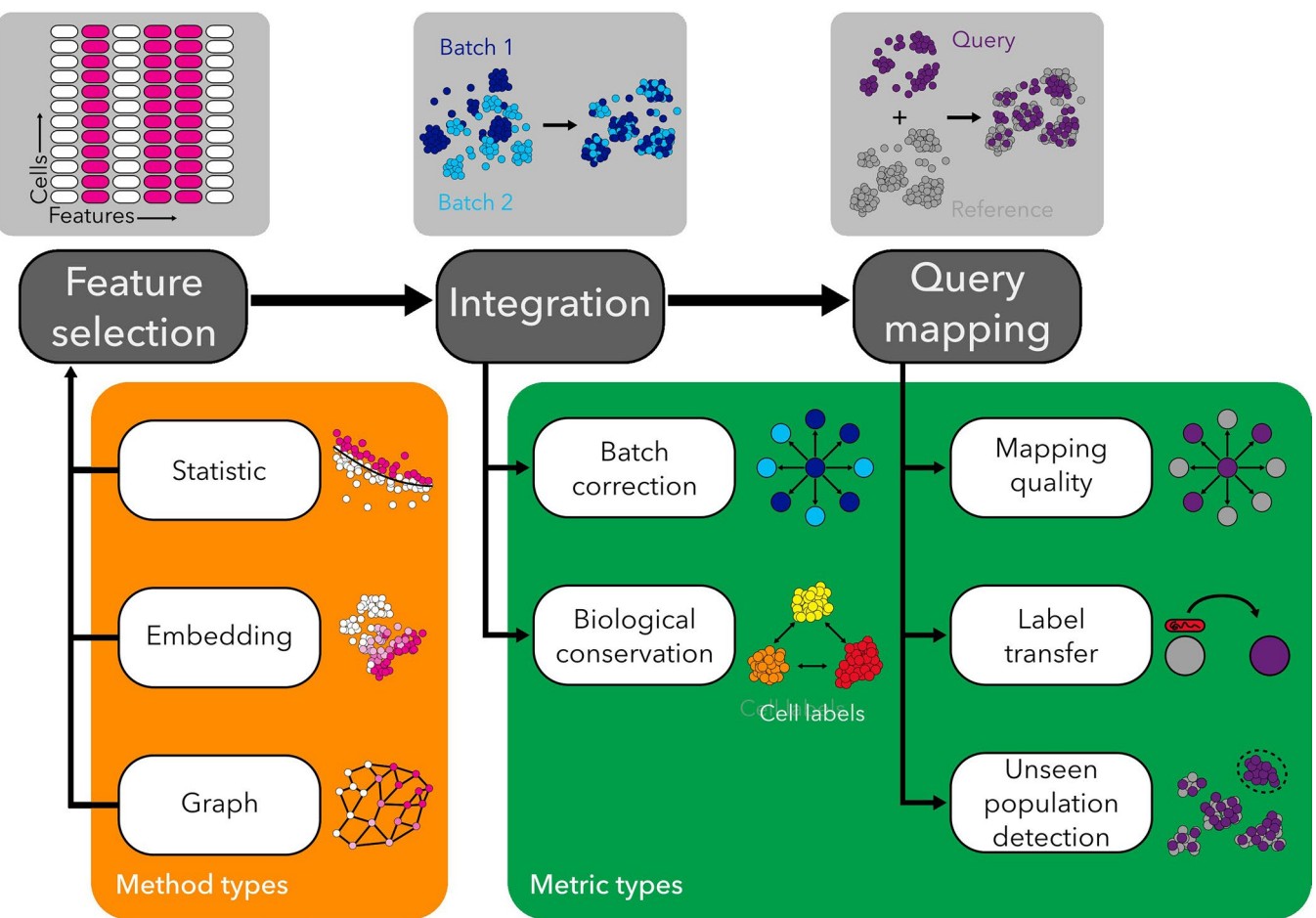

**Extended Data Fig. 1 | Overview of the design for the feature selection benchmarking study.** The methods to be evaluated are applied to each dataset and integration is performed. The query dataset is then mapped to the integrated reference. Different metrics are applied to assess batch correction, biological conservation, mapping quality, label transfer and unseen population detection.

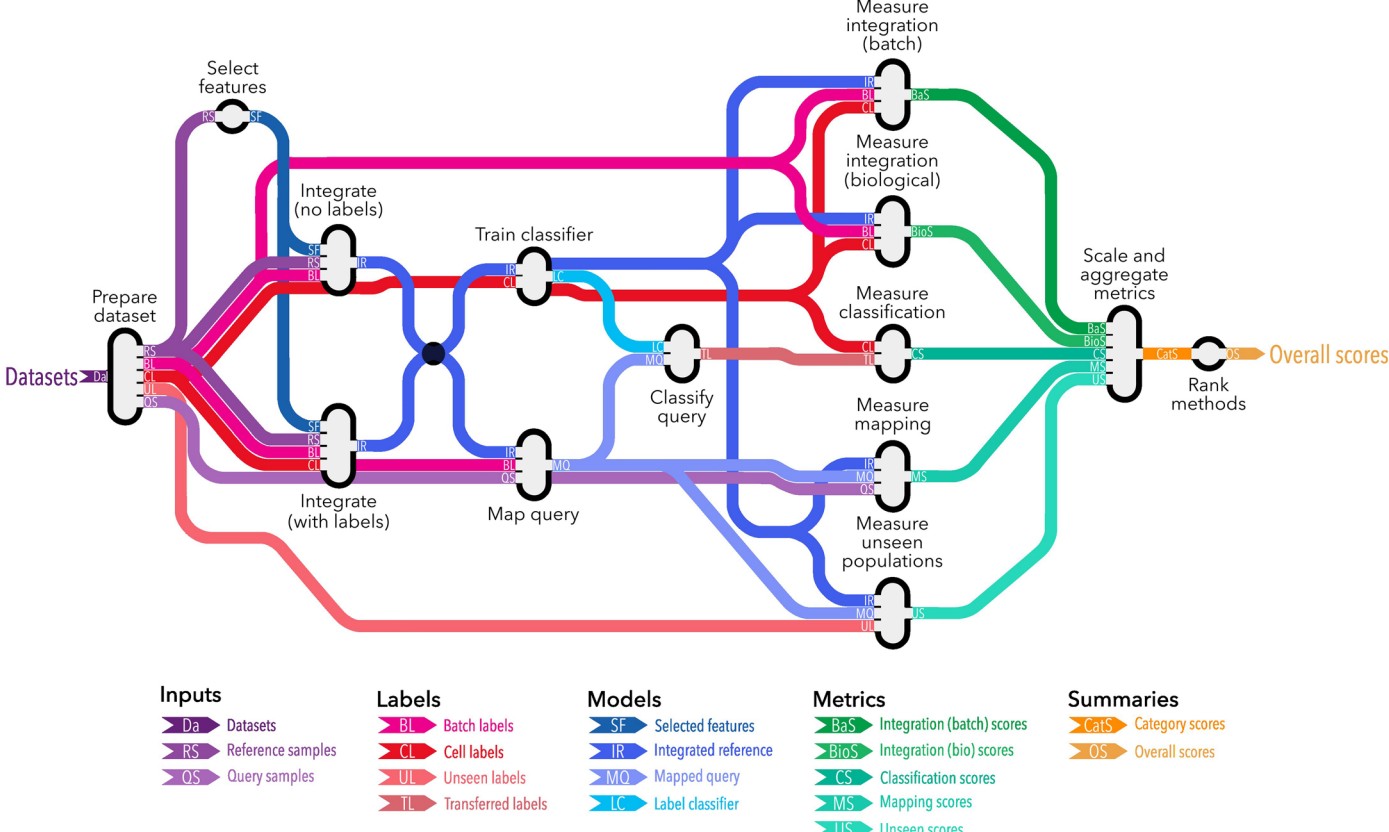

**Extended Data Fig. 2 | Schematic of the processing pipeline for the benchmark.** Light gray ovals show the processing steps and colored lines indicate the flow of information between them.

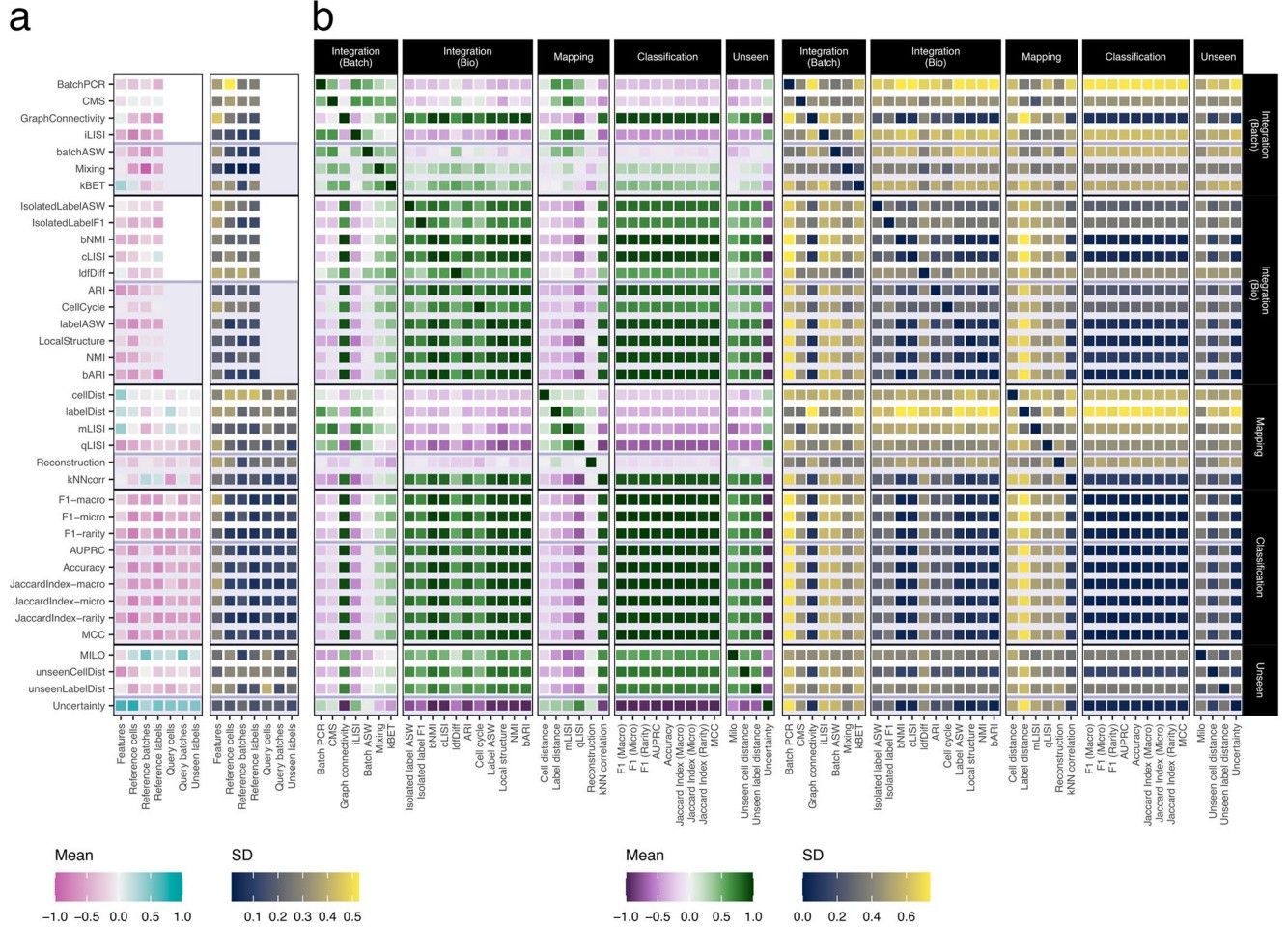

**Extended Data Fig. 3 | Metric selection correlations.** Further detail on correlations calculated during metric selection. **a**) Heatmaps of means and standard deviations for correlations between metric scores and technical dataset features. **b**) Heatmaps of means and standard deviations for correlations between metrics.

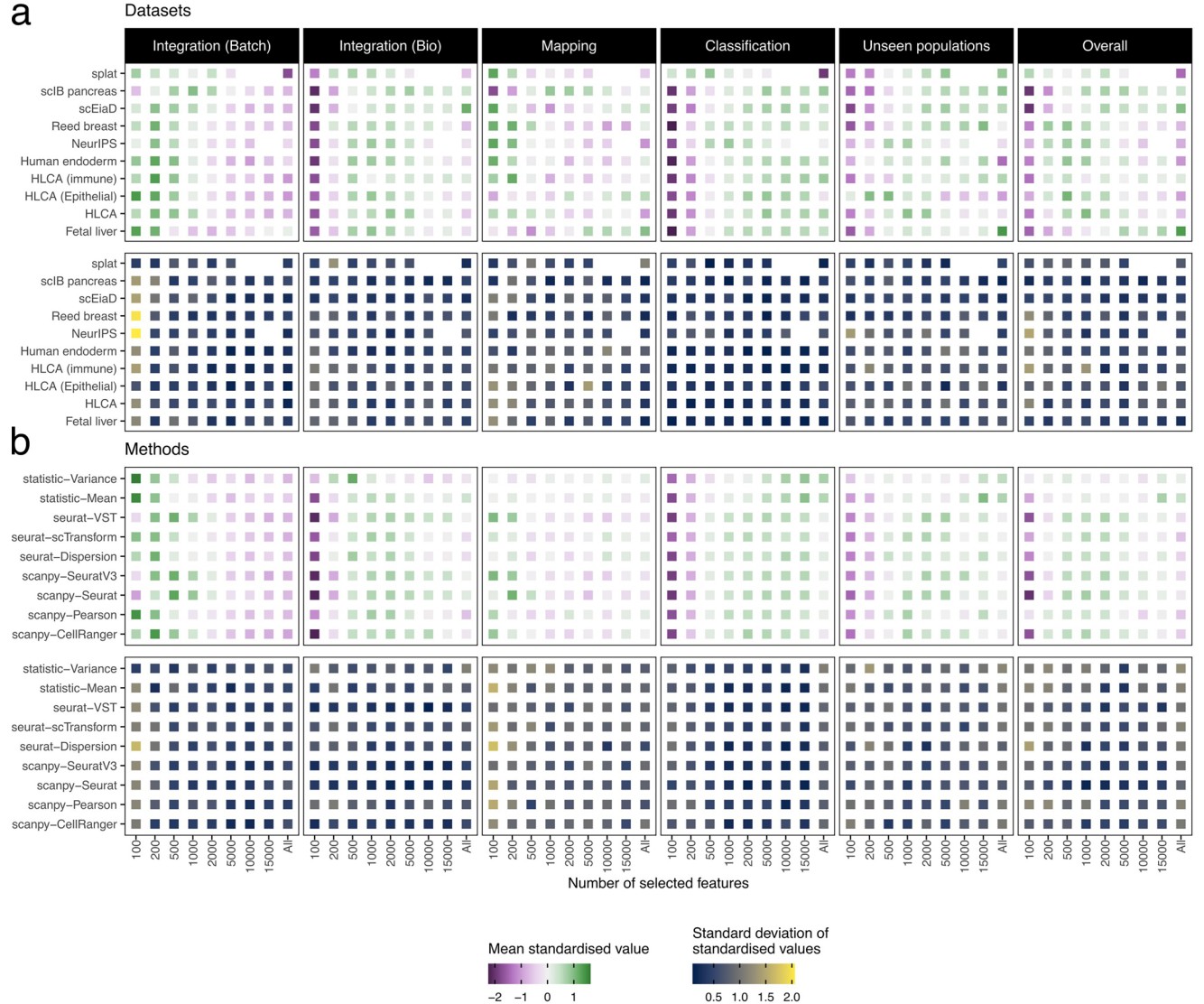

**Extended Data Fig. 4 | Metric scores for different numbers of features.** Further detail on standardized metric scores for different numbers of features. **a)** Heatmaps of means and standard deviations of standardized metric scores by metric type for different datasets and numbers of features. **b)** Heatmaps of means and standard deviations of standardized metric scores by method for different methods and numbers of features.

# a

## Metric category scores

# b

## Metric category ranks

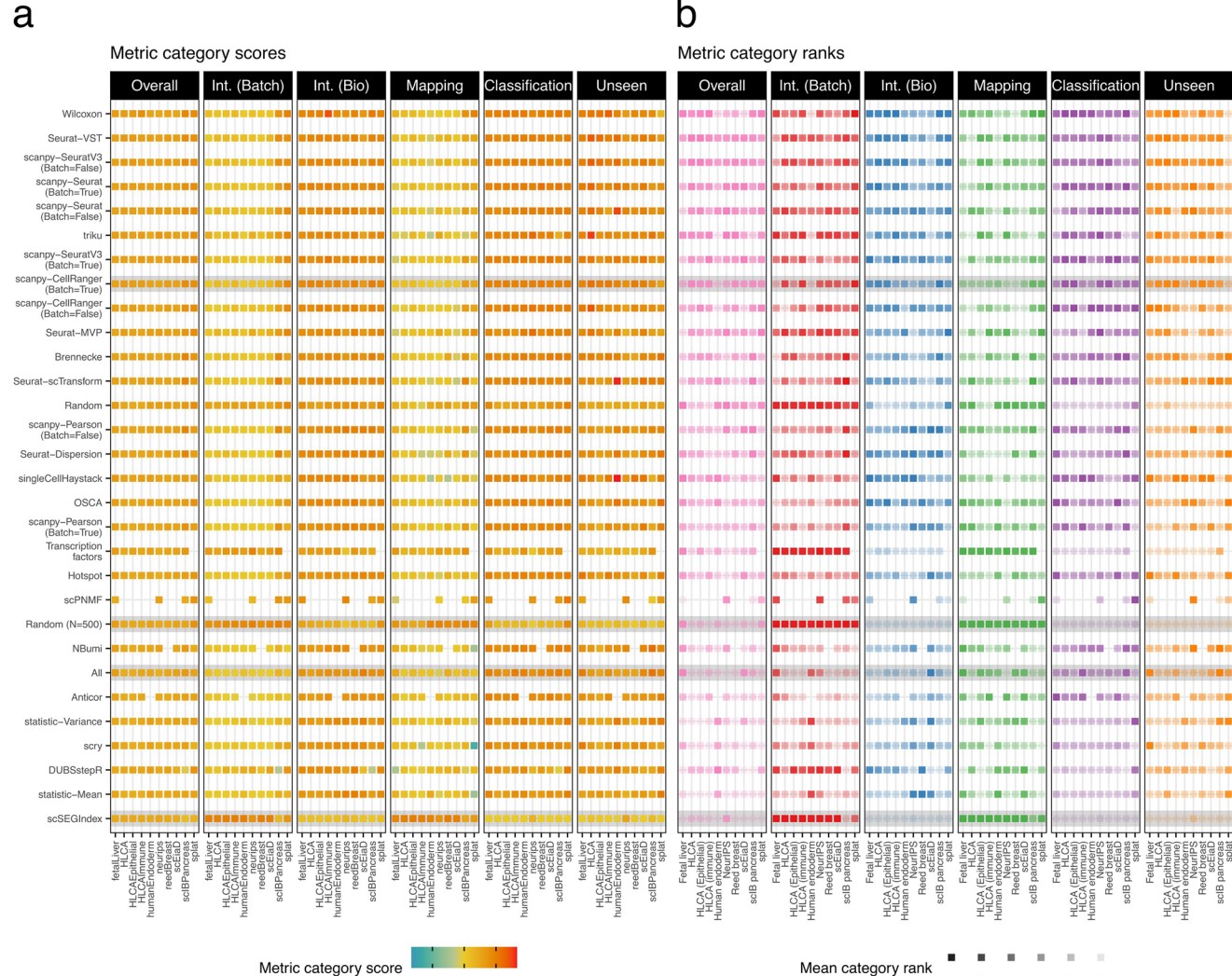

**Extended Data Fig. 5 | Benchmark metric category results.** Further detail on metric category scores and ranks for each dataset. **a**) Heatmap showing metric category scores for each method on each dataset. Colors indicate category scores. **b**) Heatmap showing metric category ranks for each method on each dataset. Colors indicate metric categories and transparency indicates rank. Baseline methods are indicated by grey shading.

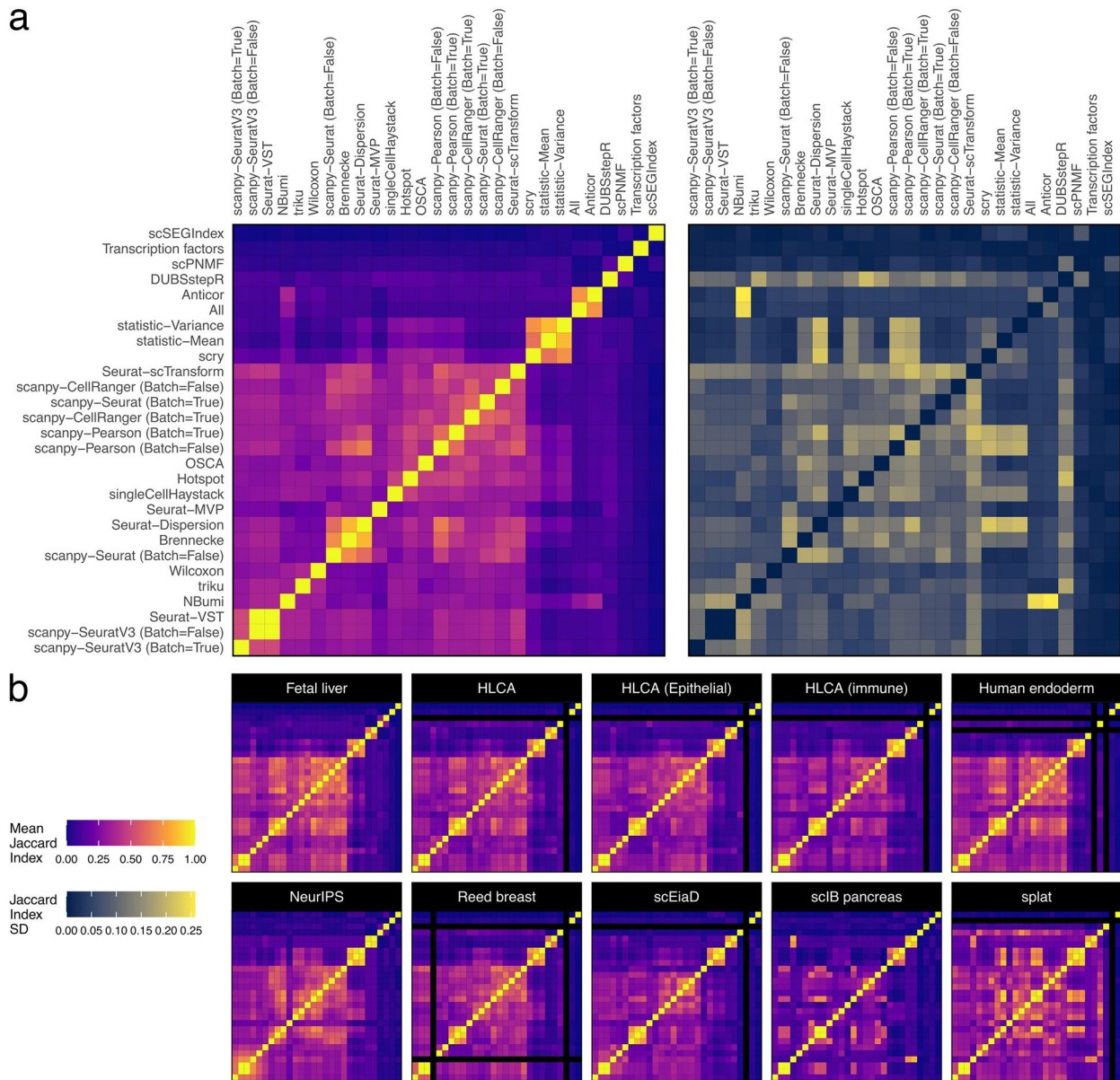

**Extended Data Fig. 6 | Selected features overlaps.** Further detail on overlaps between feature sets from different methods. **a**) Heatmaps showing mean and standard deviation of the Jaccard Index between different feature selection methods over all datasets. **b**) Heatmaps of the Jaccard Index between methods for individual datasets.

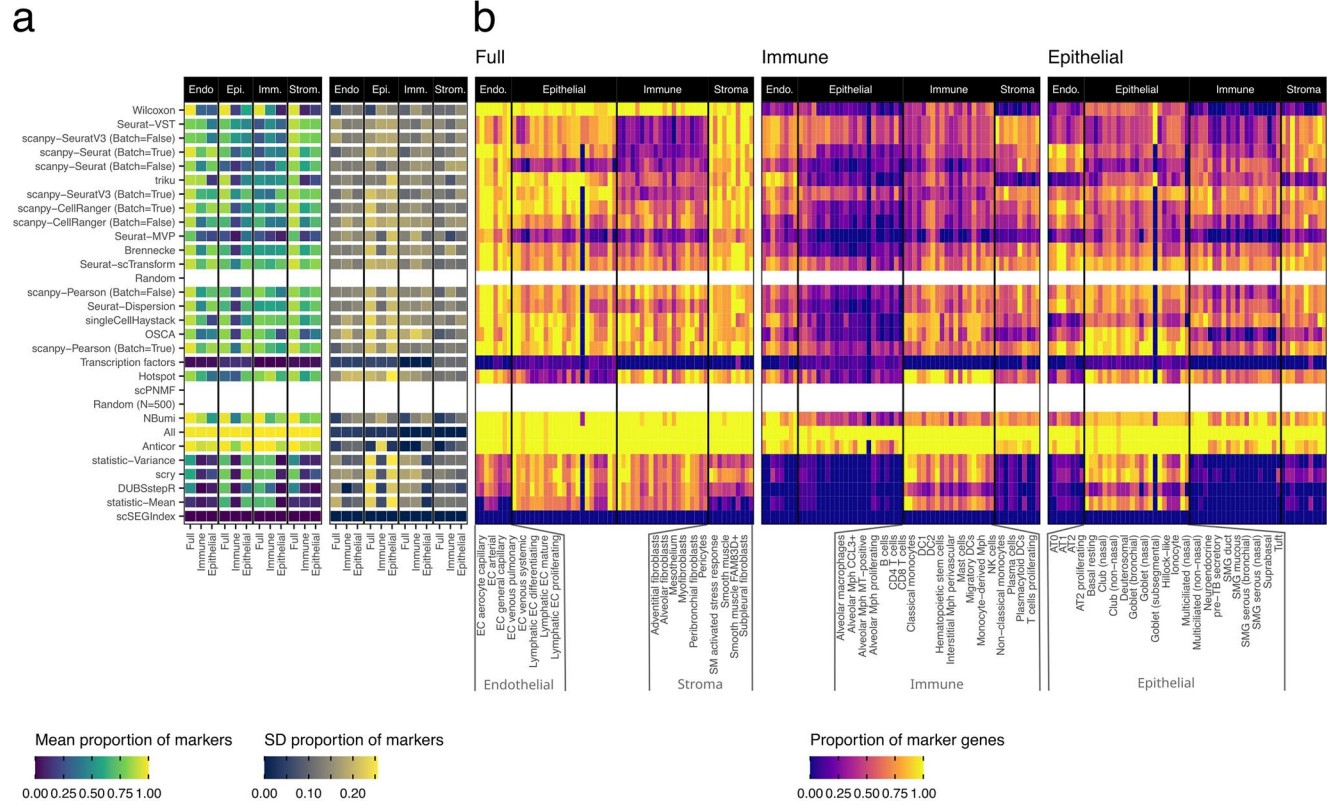

**Extended Data Fig. 7 | Marker genes overlaps.** Further detail on overlaps between selected feature sets and marker genes for HLCA datasets. **a)** Heatmaps of mean and standard deviation of the porportion of markers selection by each method on the full HLCA, HLCA (Immune) and HLCA (Epithelial) datasets for the cell types from endothelial, epithelial, immune and stroma compartments. **b)** Proportion of markers selected by methods for individual cell types on each HLCA dataset.

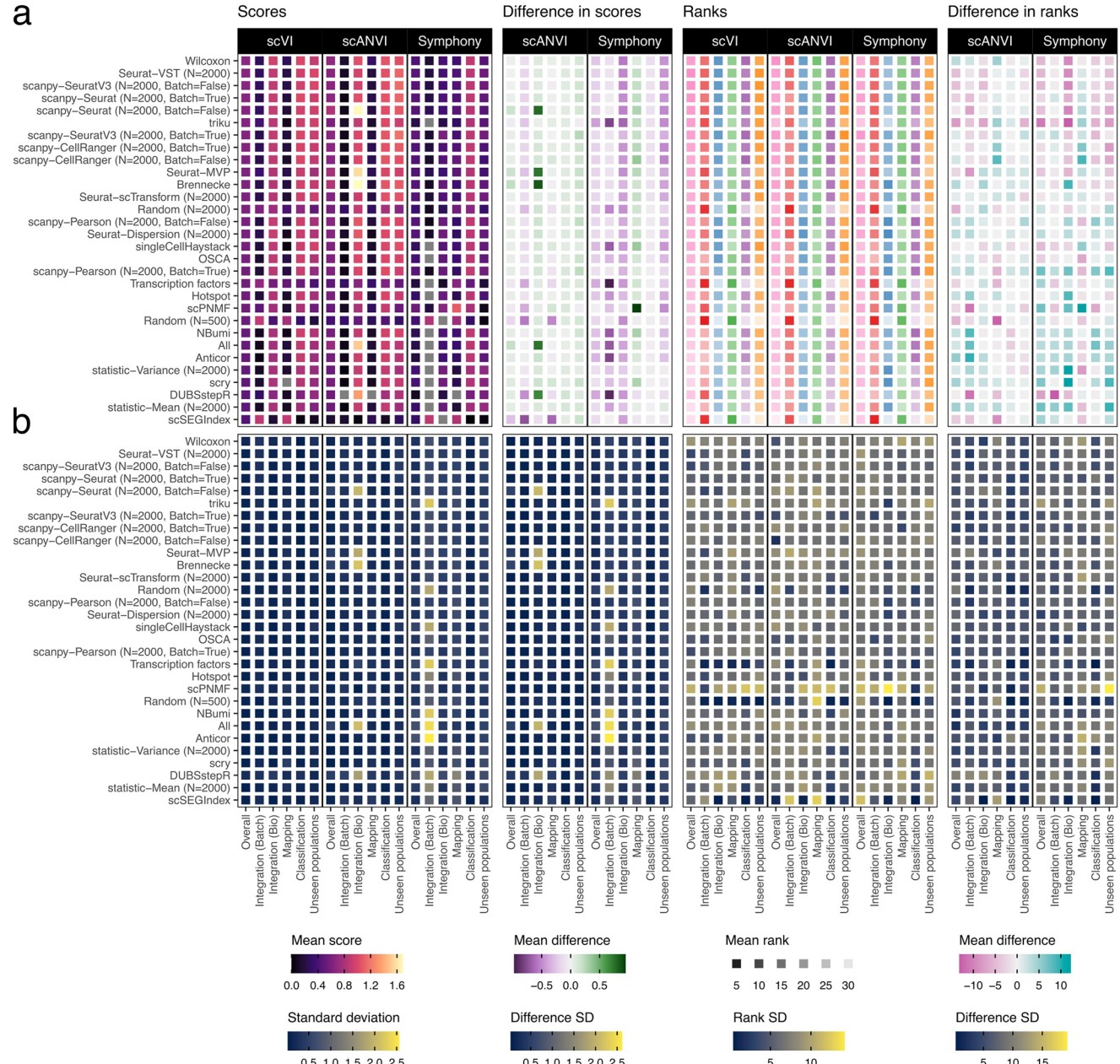

**Extended Data Fig. 8 | Integration comparison metric category results.**
Further detail on the comparison of metric category scores between integration methods (scVI, scANVI, Symphony). **a)** Heatmaps showing mean metric category scores, mean differences in scores compared to scVI, mean metric category ranks and difference in mean category ranks for each feature selection and integration method. **b)** Heatmaps showing the standard deviation in metric scores, score difference, rank and rank differences.

**Extended Data Table 1 | Summary of reasoning for excluded or modified metrics**

| Metric(s) | Type | Reasoning |
|---|---|---|
| Graph connectivity | Integration (Batch) | The correlation pattern shows greater similarity to Integration (Bio) metrics than the other Integration (Batch) metrics, so graph connectivity was included as an Integration (Bio) metric |
| Batch ASW | Integration (Batch) | Limited observed effective range |
| Mixing | Integration (Batch) | Limited effective range. Correlated with both batch and biological metrics, which confuses the signals between categories. |
| kBET | Integration (Batch) | Correlated with both batch and biological metrics, which confuses the signal between the two categories |
| ARI, NMI, bARI | Integration (Bio) | All ARI and NMI metrics showed a high correlation with each other. We selected bNMI as a representative metric. |
| Cell cycle | Integration (Bio) | Relies on the presence of cycling cells, so it is inconsistent between datasets and cannot be calculated for simulated data |
| Label ASW | Integration (Bio) | Limited observed effective range |
| Local structure | Integration (Bio) | Strong positive correlation with the number of features |
| Reconstruction | Mapping | Limited effective range, negative correlation with the number of features, inconsistent with other mapping metrics |
| kNN correlation | Mapping | Strong positive correlation with the number of features. Strong correlation with metrics from other categories. |
| AUPRC, Accuracy, JaccardIndex (Macro), JaccardIndex (Micro), JaccardIndex (Rarity), MCC | Classification | All classification metrics showed very high correlations with each other. We selected variants of F1 scores as representatives of cell-level, label-level and rarity-weighted classification metrics. |
| Uncertainty | Unseen populations | Inconsistent with other unseen metrics. Relies on the calibration of classifiers which has not been tested. |

Summary of reasoning for excluded or modified metrics

# Reporting Summary

## Statistics

For all statistical analyses, confirm that the following items are present in the figure legend, table legend, main text, or Methods section.

| n/a | Confirmed | |
|---|---|---|
| ☒ | ☐ | The exact sample size (*n*) for each experimental group/condition, given as a discrete number and unit of measurement |
| ☒ | ☐ | A statement on whether measurements were taken from distinct samples or whether the same sample was measured repeatedly |
| ☒ | ☐ | The statistical test(s) used AND whether they are one- or two-sided<br>*Only common tests should be described solely by name; describe more complex techniques in the Methods section.* |
| ☒ | ☐ | A description of all covariates tested |
| ☒ | ☐ | A description of any assumptions or corrections, such as tests of normality and adjustment for multiple comparisons |
| ☒ | ☐ | A full description of the statistical parameters including central tendency (e.g. means) or other basic estimates (e.g. regression coefficient) AND variation (e.g. standard deviation) or associated estimates of uncertainty (e.g. confidence intervals) |
| ☒ | ☐ | For null hypothesis testing, the test statistic (e.g. $F$, $t$, $r$) with confidence intervals, effect sizes, degrees of freedom and $P$ value noted<br>*Give P values as exact values whenever suitable.* |
| ☒ | ☐ | For Bayesian analysis, information on the choice of priors and Markov chain Monte Carlo settings |
| ☒ | ☐ | For hierarchical and complex designs, identification of the appropriate level for tests and full reporting of outcomes |
| ☒ | ☐ | Estimates of effect sizes (e.g. Cohen's *d*, Pearson's *r*), indicating how they were calculated |

*Our web collection on statistics for biologists contains articles on many of the points above.*

## Software and code

Policy information about availability of computer code

Data collection
> All code associated with this study is available on GitHub (https://github.com/theislab/atlas-feature-selection-benchmark/analysis) and archived on Zenodo (https://doi.org/10.5281/ZENODO.13995812), including scripts for downloading datasets from public repositories provided by the original authors, running methods and calculating metrics, the Nextflow pipeline and associated environment and configuration files. The code for analysing the benchmark results, including the production of final figures, is also available in this repository.

Data analysis
> Analysis of benchmark results was performed in R and Python using standard packages. All code is provided in the code repository on GitHub (https://github.com/theislab/atlas-feature-selection-benchmark/analysis) and archived on Zenodo (https://doi.org/10.5281/ZENODO.13995812). Rendered analysis reports including the package versions used are available from figshare (https://doi.org/10.6084/M9.FIGSHARE.C.7521966).
>
> Key package versions:
>
> anndata (v0.8.0)
> anticor (v0.1.8)
> balanced_clustering (commit: a2ae3a4d)
> batchelor (v1.14.0)
> biomaRt (v2.54.0)
> CellMixS (v1.14.0)
> cellxgene-census (v1.0.1)
> DUBStepR (commit 76aa3948)
> ggplot2 (v3.5.0)

harmonypy (v0.0.9)
hotspot (v1.0.0)
kBET (commit a10ffeaa)
m3drop (v1.24.0)
milopy (commit be1a6cc8)
numpy (v1.22.4)
pandas (v1.4.3)
patchwork (v1.2.0)
Python (v3.9.13)
R (v4.2.2)
reticulate (v1.26)
scanpy (v1.9.1)
scIB (v1.1.4)
scikit-learn (v1.1.2)
scipy (v1.9.0)
scMerge (v1.1.4.0)
scPNMF (commit 47d5b10c)
scran (v1.26.0)
scry (v1.10.0)
scTransform (v0.3.5)
scuttle (v1.8.0)
scvi-tools (v0.17.1)
Seurat (v4.3.0)
SeuratObject (v4.1.3)
SingleCellExperiment (v1.20.0)
singleCellHaystack (v0.3.4)
splatter (v1.25.1)
symphonypy (0.2.1)
tidyverse (v2.0.0)
triku (v2.1.4)
zellkonverter (v1.8.0)

For manuscripts utilizing custom algorithms or software that are central to the research but not yet described in published literature, software must be made available to editors and reviewers. We strongly encourage code deposition in a community repository (e.g. GitHub). See the Nature Portfolio guidelines for submitting code & software for further information.

# Data

Policy information about availability of data

All manuscripts must include a data availability statement. This statement should provide the following information, where applicable:
- Accession codes, unique identifiers, or web links for publicly available datasets
- A description of any restrictions on data availability
- For clinical datasets or third party data, please ensure that the statement adheres to our policy

All real scRNA-seq datasets were downloaded from public repositories provided by the original authors as described in the methods (scIB Pancreas: figshare(https://figshare.com/articles/dataset/Benchmarking_atlas-level_data_integration_in_single-cell_genomics_-_integration_task_datasets_Immune_and_pancreas_/12420968), NeurIPS: GEO (GSE194122), Fetal liver: CellAtlas.io, Reed Breast: CELLxGENE Discover (Dataset ID: 0ba636a1-4754-4786-a8be-7ab3cf760fd6, Census version: 2023-07-25), scEiaD: PLatform for Analysis of scEiad website (https://plae.nei.nih.gov/), Human endoderm: Mendelay Data (https://data.mendeley.com/datasets/x53tts3zfr/2), HLCA: CELLxGENE Discover (Dataset ID: 066943a2-fdac-4b29-b348-40cede398e4e, Census version: 2023-07-25). Raw and prepared dataset files, selected feature sets, metric scores and rendered analysis reports from this benchmark are available from figshare (https://doi.org/10.6084/M9.FIGSHARE.C.7521966).

# Research involving human participants, their data, or biological material

Policy information about studies with human participants or human data. See also policy information about sex, gender (identity/presentation), and sexual orientation and race, ethnicity and racism.

| Reporting on sex and gender | N/A |
| --- | --- |
| Reporting on race, ethnicity, or other socially relevant groupings | N/A |
| Population characteristics | N/A |
| Recruitment | N/A |
| Ethics oversight | N/A |

Note that full information on the approval of the study protocol must also be provided in the manuscript.

# Field-specific reporting

Please select the one below that is the best fit for your research. If you are not sure, read the appropriate sections before making your selection.

☒ Life sciences ☐ Behavioural & social sciences ☐ Ecological, evolutionary & environmental sciences

For a reference copy of the document with all sections, see nature.com/documents/nr-reporting-summary-flat.pdf

# Life sciences study design

All studies must disclose on these points even when the disclosure is negative.

| | |
|---|---|
| Sample size | The number of datasets in the study was chosen to cover a representative sample of scenarios and tissues as discussed and approved by reviewers in the Stage 1 review. The number of samples in each dataset was as provided by the original authors. |
| Data exclusions | For some datasets, a subset of the data was used. Either to select a relevant part of the data (such as a single tissue/species) or to for technical reasons (i.e. labels with too few cells). Any subsetting of datasets is described in that dataset's section in the manuscript. |
| Replication | Due to computational limitations the main benchmark run was performed a single time. We did not attempt to replicate the results through an independent run of the benchmarking pipeline. |
| Randomization | All correlations shown in the manuscript were performed between pairs of variables without considering other covariates. We did not perform any statistical testing for which covariates should be considered |
| Blinding | All running and scoring of methods was performed by an automated workflow. |

# Reporting for specific materials, systems and methods

We require information from authors about some types of materials, experimental systems and methods used in many studies. Here, indicate whether each material, system or method listed is relevant to your study. If you are not sure if a list item applies to your research, read the appropriate section before selecting a response.

## Materials & experimental systems

| n/a | Involved in the study |
|---|---|
| ☒ | ☐ Antibodies |
| ☒ | ☐ Eukaryotic cell lines |
| ☒ | ☐ Palaeontology and archaeology |
| ☒ | ☐ Animals and other organisms |
| ☒ | ☐ Clinical data |
| ☒ | ☐ Dual use research of concern |
| ☒ | ☐ Plants |

## Methods

| n/a | Involved in the study |
|---|---|
| ☒ | ☐ ChIP-seq |
| ☒ | ☐ Flow cytometry |
| ☒ | ☐ MRI-based neuroimaging |

## Plants

| | |
|---|---|
| Seed stocks | *Report on the source of all seed stocks or other plant material used. If applicable, state the seed stock centre and catalogue number. If plant specimens were collected from the field, describe the collection location, date and sampling procedures.* |
| Novel plant genotypes | *Describe the methods by which all novel plant genotypes were produced. This includes those generated by transgenic approaches, gene editing, chemical/radiation-based mutagenesis and hybridization. For transgenic lines, describe the transformation method, the number of independent lines analyzed and the generation upon which experiments were performed. For gene-edited lines, describe the editor used, the endogenous sequence targeted for editing, the targeting guide RNA sequence (if applicable) and how the editor was applied.* |
| Authentication | *Describe any authentication procedures for each seed stock used or novel genotype generated. Describe any experiments used to assess the effect of a mutation and, where applicable, how potential secondary effects (e.g. second site T-DNA insertions, mosiacism, off-target gene editing) were examined.* |

