## [Peer Review File · Nature Methods]

Feature selection methods affect the performance of scRNA-seq data integration and querying

Corresponding Author: Professor Fabian Theis

Version 0:

Decision Letter:

16th Dec 2022

Dear Professor Theis,

Thank you for your inquiry about submitting your manuscript, "Feature selection methods affect the performance of scRNA-seq data integration and querying" to Nature Methods. The paper sounds like it may be of interest, and should fit the scope of the journal. We would be willing to consider it for publication in Nature Methods as a Registered Report.

Of course, it is very difficult to judge a paper based only on the limited information available in a presubmission inquiry. Therefore I am sure you understand that we cannot promise to send your paper out for peer review and must read it in its entirety before deciding if this would be suitable.

Our Registered Reports format is intended for systematic performance comparison studies of related, established methods or tools. We strongly encourage you to read our [Editorial](https://www.nature.com/articles/s41592-022-01407-4) introducing this format and familiarize yourself with our [guidelines](https://www.nature.com/nmeth/submission-guidelines/registered-reports) for this format. As the purpose of a Registered Report is to provide authors with editor and reviewer feedback at the experimental design phase, Stage 1 manuscripts should be submitted prior to embarking on large-scale data collection (pilot data is fine and encouraged).

You will find our Guide to Authors at <http://www.nature.com/naturemethods> to assist you in preparing your manuscript. However, it is not necessary at this stage to spend major effort adhering to our detailed formatting instructions.

Thank you for your interest in Nature Methods.

Sincerely yours,

Allison Doerr, Ph.D.
Chief Editor
Nature Methods

Version 1:

Decision Letter:

30th Aug 2023

Dear Fabian,

Please accept our sincere apologies for the length of time that it has taken to get this decision to you. Your Registered Report Stage 1 proposal, "Feature selection methods affect the performance of scRNA-seq data integration and querying", has now been evaluated by 4 reviewers. As you will see from their comments below, although the reviewers find your work of considerable potential interest, they have raised several suggestions and concerns. We are very interested in the possibility of proceeding further with your submission at Nature Methods, but would like to consider your response to these concerns before

we reach a final decision on acceptance in principle and Stage 2 submission.

I have discussed the reviewer reports in detail with our chief editor and other members of the Nature Methods team, with a view to 1) identifying key priorities that should be addressed in revision, and 2) overruling reviewer requests that we think fall beyond the scope of the study. Please see the attached file which contains the reviewer reports annotated with our editorial recommendations. We invite you to revise your Stage 1 proposal to address these concerns (before you embark on the proposed experiments).

We are committed to providing a fair and constructive peer review process. Do not hesitate to contact us if there are any reviewer requests that you believe are unlikely to yield a meaningful outcome.

- * include a point-by-point response to the reviewers and to any editorial suggestions
- * please underline/highlight any additions to the text or areas with other significant changes to facilitate review of the revised manuscript
- * include a cover letter with the following information

- an anticipated timeline for completing the study (should the study be accepted in principle)
- a statement confirming that, should the study be accepted in principle, you agree to share your raw data and conform to our other open science requirements (see details below)
- a statement confirming that, following acceptance in principle, you agree to register the Stage 1 proposal in a recognized repository (either publicly or under private embargo), until submission of the Stage 2 manuscript (please note we can assist you in uploading your Stage 1 proposal in our https://springernature.figshare.com/registered-reports_nmethods Figshare space)
- a statement confirming that if you later decide to withdraw your manuscript from consideration at Nature Methods, you agree to the journal publishing a brief note about the withdrawn proposal

Please resubmit all the necessary files electronically by using the link below to access your home page:

Link Redacted

IMPORTANT: This URL links to your confidential home page and associated information about manuscripts you may have submitted, or that you are reviewing for us. If you wish to forward this email to co-authors, please delete the link to your homepage.

We hope to receive your revised proposal within 12 weeks. If you cannot send it within this time, please let us know. In this event, we will still be happy to reconsider your paper at a later date so long as nothing similar has been accepted for publication at Nature Methods or published elsewhere.

ORCID

Nature Methods is committed to improving transparency in authorship. As part of our efforts in this direction, we are now requesting that all authors identified as 'corresponding author' on published papers create and link their Open Researcher and Contributor Identifier (ORCID) with their account on the Manuscript Tracking System (MTS), prior to acceptance. This applies to primary research papers only. ORCID helps the scientific community achieve unambiguous attribution of all scholarly contributions. You can create and link your ORCID from the home page of the MTS by clicking on 'Modify my Springer Nature account'. For more information please visit <http://www.springernature.com/orcid>.

Best regards,
Lei

Lei Tang, Ph.D.
Senior Editor
Nature Methods

Reviewers' Comments:

Reviewer #1:

Remarks to the Author:

I have reviewed the proposed benchmarking paper according to the criteria provided by the editor.

1) The importance of the comparative analysis and relevance for a broad, multidisciplinary audience.

The ultimate goal of this manuscript is to improve single cell data integration and querying, two important tasks in the broader field of single cell data analysis. Given the importance of single cell technologies in biology and medicine, the notion of improving data integration and querying is clearly significant.

2) The rationale, plausibility and comprehensiveness of the proposed comparative analysis.

This manuscript specifically proposes to benchmark feature selection methods in the context of single cell data integration and querying. Feature selection is a critical step in data preprocessing for single cell data analysis, as *all* downstream analysis tasks (visualization, clustering, integration, querying, etc.) operated on the selected feature set. This study therefore has strong rationale.

The proposed study as is, is also plausible -- the senior authors have executed a very similar benchmarking study on integration methods previously, and are building on their previously successful benchmarking work.

I have strong issues with the comprehensiveness of this study as designed, however. Central to the problems of integration and querying is the notion that cell populations/types have varying levels of heterogeneity driven by both biological factors (e.g. conditions) and technical factors (batch, etc.), as pointed out by the authors. Feature selection plays an important role in integration in part because it defines the input features which are used both to find matching cell types across batches, and whose variation it is assumed to capture the biological variability within a population. One can therefore easily imagine that the "optimal" set of input features is a function of cell types present, organ systems, technology, etc. A large blindspot for the current benchmark proposal is studying what the effect of these important biological factors (cell type, organ, technology, species) is on both the 'performance' of the feature selection methods, and whether certain methods are better suited to capture features for certain cell types/systems/etc. than others. As the senior authors are obviously aware given their work on HLCA and the HCA in general, there are huge atlases from many different species that are generated from multiple groups, technologies, etc., that could be leveraged to study the effect of key biological factors like cell type on feature selection. That would, in my mind, increase the value of the benchmark paper significantly. While the authors propose to use 8 different datasets for the benchmarking, it would be much more informative if the authors also chose bigger datasets like from the HCA or Tabula muris/sapiens, where they could systematically vary biological factors like cell type (without jumping across different datasets to do so). Such an analysis would then allow the authors to provide more generalizable recommendations for how to select the best feature selection approach for their own work.

I am also *very* surprised the manuscript doesn't propose to vary the number of features selected (given the same feature selection approach) -- as an end user, it is completely unclear how the number of features selected is chosen (particularly for methods where that choice must be made explicitly by the user). It is extremely odd for the proposed benchmark to actively decide not to vary this for each chosen method, given it is a benchmarking paper about feature selection. This needs to be rectified by the authors to be a valuable study to the community.

3) The soundness and feasibility of the methodology and data analysis pipeline (including statistical power analysis where appropriate).

Given the caveats mentioned above regarding lack of comprehensive exploration of the dependence of results on cell types / organ systems / etc. mentioned above, the proposed evaluation metrics, reusable workflows, etc., are sound and feasible, particularly given the senior authors' past work on benchmarking single cell integration methods.

4) Whether the clarity and degree of methodological detail would be sufficient to replicate exactly the proposed experimental procedures and data analysis pipeline.

I have no concerns on replicability of the work - the senior authors have demonstrated from their past work that they generate reproducible benchmarking studies.

5) Whether the proposed comparative analysis is subject to any bias.

None that I can see.

Reviewer #2:

Remarks to the Author:

Comments to authors

Zappia et al. present a plan to perform a bench-marking analysis of feature selection for scRNA-seq, and assess the impact on data integration and reference-query mapping. They provide a comprehensive list of potential metrics, and importantly, include plans for a comparative analysis of such metrics to select the "most relevant". Broadly, the plan is thorough and has the potential to greatly inform both standard practises in single-cell atlas building, query-reference mapping, as a reference for new-comers to scRNA-seq analysis to select an appropriate feature selection algorithm.

That said, there is room for clarification, and some adjustments to their current strategy are recommended.

Many of the evaluation metrics require a ground truth, which the authors here define as the reference data in the reference-query problem setting. However, this approach assumes the reference is unbiased (v.strong assumption), and so opens the door to the metrics also being biased. The benchmarking strategy would greatly benefit from simulations and/or synthetic data to define ground truths; the latter are suggested due to the general limitations on realism for simulated single-cell gene

expression data on the scale and complexity of atlas data sets. The inclusion of simulations, and then comparing the simulation vs. real data vs. synthetic data sets will provide a good indicator of the consistency of the benchmarking metrics. If they are inconsistent, then it shows that either the simulations were unrealistic (need better simulations) or that the metrics or feature selection methods are not consistent.

Is the expectation that a random set of features would be less performant than the feature selection methods a reasonable one? Random subsets of genes might perform surprisingly well as they will capture a lot of structure based on gene-gene correlations. Particularly as the random subsets would need to be matched to selected features on composition, such as gene expression level distributions, sequence-ability, etc, to be a fair null comparison. When comparing to random subsets, how many permutations of the random feature set are used? This doesn't seem to be defined and is implicitly = 1. This could be limited, and prone to sampling variation. Moreover, does a random subset need to be computed for each method to provide a valid null comparison?

It is not clear why different implementations of the same method would be required. Unless of course these are not implemented the same. In which case, they should not be labelled as the same method and divergence between algorithms should be clearly stated for all users, especially newcomers to the field. The justification for their inclusion based on pre-processing differences seems rather weak, as the focus is then on the pre-processing, and is not related to the feature selection algorithms/methods per se.

The focus is largely on data sets generated by drop-based scRNA-seq chemistries. While this may be the chemistry du jour, inclusion of legacy and alternative chemistries warrants inclusion. Especially as future atlas/meta-analyses will make use of publicly available data.

For several non-excessive variability methods, an initial feature set is required. Across these methods, inconsistencies arise in the initial feature selection method, which could be a huge source of (a) variation and (b) bias. Instead of the current categorisation, methods could be split into primary and secondary methods, with the secondary methods are applied to the out from the primaries to give a fairer comparison. The natural cost is many more comparisons, but this strikes me as being considerably fairer and less prone to bias and inconsistencies.

The supervised methods rely on published methods, but omits biologically motivated feature sets. For instance, all transcription factors, or all cell surface expressed genes, as these may also be highly characteristic of the cell type diversity in an atlas/sample.

For each evaluation metric, a short explanation of whether a low or high score is better/worse would greatly benefit readers not familiar with the technicalities of each evaluation method.

A more principled selection of metrics would be most informative for readers and for the research team in terms of interpreting results and making recommendations. This indicates a pilot experiment to select informative and non-redundant metrics is required before undertaking the full benchmarking. While the authors propose to perform a 'metric selection analysis' there is no details on how this will be achieved and is one of the biggest omissions of the study design. If anything, this metric selection process should have already been performed.

An explanation of the assumptions of each method and evaluation metric would greatly benefit the interpretation for the reader. More concrete details are required for many metrics, for instance, qualitative thresholds such as "fewer" are ill-defined and should be made explicit.

For several evaluation metric, it is not clear what the random distribution of values would look like. While the pilot data attempts to show this I am not convinced the "null" methods used are truly "null" in the sense that the feature sets will be depleted for "true" cell identity/state/etc defining gene sets.

Presentation of analyses: The central tendency and the spread of values should be reported. e.g. 2 methods may have moderate scores across metrics, but 1 with a smaller spread than the other. The smaller spread would imply greater consistency across measures, and thus more confidence in its ranking compared to the method with a large spread.

Minor/method/metric-specific comments

The authors refer to "flavours" in the Seurat and cellranger excessive variability methods. Is this a typographical error (function?) or the improper use of a colloquialism?

For the NBumi method the authors state they will select features with $p < 0.01$ or the 500 lowest p-values, which ever gives the larger feature set. This could be a source of bias, and should be justified with e.g., a sensitivity analysis, especially as most of the metrics/assessment is performed down-stream of feature selection, i.e., in terms of reference-query mapping.

How does the scanty implementation of the Pearson residuals method differ from that of the original by the Irizarry group? Why is the original not included?

For the singleCellHaystack method the authors state that they will use the Seurat HVG method for the initial round of PCA. Does this just return a feature set that is strongly enriched for the original set? Is this tautological? Should this use each of the different feature selection methods for the initial feature selection step to ensure that it doesn't just recapitulate the rankings of all other methods?

For the Hotspot method, as with singleCellHaystack, this has the potential for a circular result. The same applies to triku – what feature set is used for the initial PCA?

For the Batch PCR evaluation are the regression coefficients standardised? Are the components matched, i.e. does PC1 correspond to integrated dimension 1? Or is this the sum over all coefficient-weighted variances? This will show that variance is removed and from which components. Should the test be on a component-wise basis to demonstrate that non-batch axes of variance are preserved? More details required here.

For iLISI how sensitive is this the proposed approach to the number of original batches? What is the null distribution, a multinomial (binomial for 2 batches?)

For the CMS tests, does the implementation assume a specific distribution, or is it undefined as in the K-sample Anderson-Darling test?

The statement that “The scanpy methods is ...an example of the currently accepted best practise...” – I would like to see a citation or evidence to support this statement.

Reviewer #3:

Remarks to the Author:

The proposed study aims to evaluate the effects of feature selection on single-cell RNA-seq integration (batch correction) and query mapping with label transfer. Overall, the study seems carefully designed and comprehensive and the findings would be useful to the single-cell community.

My main question/concern has to do with the limited choice of integration method used for testing (scVI and it's extension scANVI), I believe this analysis would be much more powerful if it considered a handful (even just 1-2) of other commonly used integration methods – a recent benchmarking of integration methods coming from the same group, for example (manuscript ref 3) found Scanorama and fastMNN to be among the top performing methods. Anecdotaly, variants of MNN and Harmony among others are still being commonly used in single-cell analyses. My suggestion would be to include another integration method in this analysis to widen the relevance and applicability.

Additionally, I have a few minor notes:

- For the method singleCellHaystack, why do you plan to use Seurat defined HVGs for the PCA? Why not run PCA with all features so as to not to bias the chosen features?

- It would be useful for the authors to note why they are choosing each of the query sets. In some cases it seems obvious (when there are only two major batch types for example), but in others the choice seems arbitrary – would you expect any difference in results if another outgroup was chosen?

Reviewer #4:

Remarks to the Author:

Luke Zappia et al. benchmarked gene selection methods for integration and reference usage on single-cell data. Although this study evaluated various datasets with different feature selection methods/algorithms and various metrics (integration-batch effect, integration-biological conservation, mapping quality, label transfer, and unseen population prediction), I have several questions and concerns about this manuscript, which I think are important to be addressed.

Major comments

1. The title of the study said “Feature selection methods affect the performance of scRNA-seq data integration and querying”. However, the authors only focus on deep learning integration methods. I think the author should include other popular integration methods, such as scGen, Harmony, Seurat, fastMNN, Scanorama, and BBKNN. All these methods performed well in one of your previous articles (Malte D. Luecken et al, 2022, Nature Methods).

2. Feature selection is indeed a fundamental step in single-cell RNA sequencing (scRNA-seq) data analysis. However, the significance of comparative analysis on this topic has been somewhat diminished by the publication of (Malte D. Luecken et al, 2022, Nature Methods)

3. In real data analysis, we will have different focuses for different tasks. When evaluating different feature selection methods, the author should provide recommendations on which HVG selection methods are best for different tasks instead of simply giving an overall score.

4. Although the author used various feature selection methods, it is important to know how many genes are selected in

common or differently with different methods. The author should provide a detailed table or figure for each dataset to illustrate this.

5. I think most benchmark datasets were likely annotated using the default feature selection methods in tools like Seurat or Scanpy. Therefore, it is important for authors to consider potential systemic bias within these benchmark datasets.

6. For the MILO metric, the author should give a clear explanation of why this method can be used as a metric for identifying unseen populations and should specify this metric calculated at the cellular level or at the category level. In addition, the author only used a subset of cells (up to 20,000 cells or 10 percent of the datasets, whichever is higher) and used five times the number of batches neighbors (up to a maximum of 200), do these changes have an impact on the evaluation of the method?

7. The tumor microenvironment (TME) exhibits apparent heterogeneity in different types of cancer. I recommend that authors should utilize multiple datasets about tumor TME to avoid possible evaluation bias from a single dataset.

Minor comments:

1. "Many computational scientists have tackled the integration problem using a variety of methods and there are now at least 200 tools for single-cell integration available", which cited Ref2, but Ref2 is an article published in 2018. However, most integration methods for single-cell data were developed after 2018, so the authors should check the cited references.

2. For the metric cell distance in mapping quality, how do you convert distance into the p-value of the chi-squared distribution, and what is the rationale behind this approach? And I think the metric cell distance in mapping quality is redundant.

3. All metrics the author used are normalized to [0,1], but why do some methods have negative values in their metrics?

4. There was an article on the evaluation of tools for highly variable gene discovery from single-cell RNA-seq data (PMID: 29481632), but the authors missed it.

OPEN SCIENCE REQUIREMENTS

REPORTING SUMMARY AND EDITORIAL POLICY CHECKLISTS

IMAGE INTEGRITY

DATA AVAILABILITY

All novel DNA and RNA sequencing data, protein sequences, genetic polymorphisms, linked genotype and phenotype data, gene expression data, macromolecular structures, and proteomics data must be deposited in a publicly accessible database, and accession codes and associated hyperlinks must be provided in the "Data Availability" section.

CODE AVAILABILITY

Please include a "Code Availability" subsection in the Online Methods which details how your custom code is made available. Only in rare cases (where code is not central to the main conclusions of the paper) is the statement "available upon request" allowed (and reasons should be specified).

MATERIALS AVAILABILITY

Version 2:

Decision Letter:

15th Dec 2023

Dear Fabian,

Thank you once again for submitting your revised Stage 1 Registered Report, entitled "Feature selection methods affect the performance of scRNA-seq data integration and querying" and sending the letter that responds to reviewer #4's final comments. I am delighted to say that we can offer acceptance in principle. You may progress to Stage 2 and complete your study as approved.

As you know, a condition of acceptance-in-principle is that the authors agree to deposit their Stage 1 accepted manuscript in a repository, either publicly or under embargo until Stage 2 manuscript acceptance and publication. We are very keen to showcase our accepted-in-principle manuscripts, so that our readers, reviewers, and potential authors can gain insight into the requirements of the format as well as an idea of the types of projects that are suitable for publication as Registered Reports in Nature Methods. We have set up a space on figshare (https://springernature.figshare.com/registered-reports_nmmethods) to host all of our accepted-in-principle manuscripts, which can either be made public or kept under embargo until Stage 2 acceptance (depending on author preference). This gives you the opportunity to have your work publicly associated with Nature Methods, and of course we will be very pleased to showcase your report if you agree to share it publicly.

If you agree with posting your Stage 1 manuscript on our figshare space, we will upload it (the version incorporated the changes discussed in your recent letter) on your behalf and either set it public or place it under embargo, depending on your choice. Your protocol will be licensed under a CC BY license (Creative Commons Attribution 4.0 International License). The CC BY license allows for maximum dissemination and re-use of open access materials and is preferred by many research funding bodies. Under this license users are free to share (copy, distribute and transmit) and remix (adapt) the contribution including for commercial purposes, providing they attribute the contribution in the manner specified by the author or licensor (read full legal code: <http://creativecommons.org/licenses/by/4.0/legalcode>) Please note that any use of <https://springernature.figshare.com> will be subject to the figshare terms of use. Figshare has the right to enforce these terms and conditions where applicable. Use of third party services and sites will be subject to the relevant terms of use and will apply if we act on your behalf in this regard. Do let me know if you would like to take up this option or if you have any questions regarding the manuscript deposition requirement.

Please also note that depositing the work on our figshare space does not preclude deposition of your Stage 1 manuscript on other repositories – your manuscript can also be posted on any other public repository of your choice.

Following completion of your study, we invite you to resubmit your finalized manuscript as a Stage 2 Registered Report. We will send the Stage 2 manuscript to our reviewers for a final check, but they will be instructed that any comments on novelty and/or potential significance of the results will not factor into our final decision.

IMPORTANT: Please note that your manuscript can still be rejected for publication at Stage 2 if the Editors consider any of the following to hold:

- The authors substantially alter the rationale for the study as approved in the Stage 1 submission (please note that the Introduction should not be significantly modified from the Stage 1 manuscript).
- The authors fail to adhere closely to the approved experimental plan. (Please contact us as soon as possible for advice if at any point you need to make any changes to your experimental plan.)
- The authors' conclusions are not justified given the data obtained.
- Any post hoc (unregistered) analyses are not justified or are overly dominant in shaping the authors' conclusions.
- Our open science requirements (detailed below) are not followed.

Should authors choose to withdraw their Registered Report at any time, we will publish a Withdrawn Registration notice.

When you are ready, please use the following link to access your home page and submit your Stage 2 Registered Report:

Link Redacted

We expect your Stage 2 Registered Report to be submitted by the date specified in your latest cover letter. If unforeseen circumstances prevent submission by that date, please contact us as soon as possible.

Please let me know if you agree with posting your Stage 1 manuscript on our figshare space. We will upload it on your behalf and either set it public or place it under embargo, depending on your choice.

Thank you again for submitting your work to Nature Methods and we look forward to receiving your Stage 2 Registered Report! Please do not hesitate to reach out to me at any time if you have questions.

Yours sincerely,
Lei

Lei Tang, Ph.D.
Senior Editor
Nature Methods

OPEN SCIENCE REQUIREMENTS

REPORTING SUMMARY AND EDITORIAL POLICY CHECKLISTS

When submitting your Stage 2 manuscript, please include a reporting summary and editorial policy checklists. If you have any questions about these checklists, please see <http://www.nature.com/authors/policies/availability.html> or contact me.

IMAGE INTEGRITY

DATA AVAILABILITY

All novel DNA and RNA sequencing data, protein sequences, genetic polymorphisms, linked genotype and phenotype data, gene expression data, macromolecular structures, and proteomics data must be deposited in a publicly accessible database, and accession codes and associated hyperlinks must be provided in the "Data Availability" section.

CODE AVAILABILITY

Please include a "Code Availability" subsection in the Online Methods which details how any custom code is made available. Only in rare cases (where code is not central to the main conclusions of the paper) is the statement "available upon request" allowed (and reasons should be specified).

MATERIALS AVAILABILITY

More details about our materials availability policy [can be found here](https://www.nature.com/nature-portfolio/editorial-policies/reporting-standards#availability-of-materials).

ORCID

TRANSPARENT PEER REVIEW

Nature Methods offers a transparent peer review option for new original research manuscripts submitted from 17th February 2021. We encourage increased transparency in peer review by publishing the reviewer comments, author rebuttal letters and editorial decision letters if the authors agree. Such peer review material is made available as a supplementary peer review file.

Please state in the cover letter 'I wish to participate in transparent peer review' if you want to opt in, or 'I do not wish to participate in transparent peer review' if you don't. Failure to state your preference will result in delays in accepting your manuscript for publication.

Version 3:

Decision Letter:

Our ref: NMETH-RR51186C

22nd Sep 2024

Dear Dr. Theis,

Thank you for submitting your Stage 2 Registered Reports manuscript "Feature selection methods affect the performance of scRNA-seq data integration and querying" (NMETH-RR51186C). It has now been seen by the original referees and their comments are below. In light of the reviewers' comments, we'll be happy in principle to publish the manuscript in Nature Methods, pending minor revisions to satisfy the referees' final requests and to comply with our editorial and formatting guidelines.

TRANSPARENT PEER REVIEW

Nature Methods offers a transparent peer review option for new original research manuscripts submitted from 17th February 2021. We encourage increased transparency in peer review by publishing the reviewer comments, author rebuttal letters and editorial decision letters if the authors agree. Such peer review material is made available as a supplementary peer review file.

Please state in the cover letter 'I wish to participate in transparent peer review' if you want to opt in, or 'I do not wish to participate in transparent peer review' if you don't. Failure to state your preference will result in delays in accepting your manuscript for publication.

ORCID

Sincerely,
Lei

Lei Tang, Ph.D.
Senior Editor
Nature Methods

Reviewer #2 (Remarks to the Author):

The authors have stuck to their registered analysis plan, and have not chosen to focus overly on any additional exploratory analyses. The analysis of batch-aware methods is an addition from the original analysis plan, and is well justified.

Overall, the authors have stuck closely to their registered analysis and executed it well - the presentation is generally very clear, and the authors should be commended for undertaking this important work for the single-cell research community. Figure 3 is especially clear, and informative. It provides a strong justification and important context for users when choosing a feature selection method.

I have some minor comments regarding presentation of results, and suggestions to improve clarity.

In the second paragraph of the results "Metric selection is key for reliable benchmarking", it would help the reader if the authors could state explicitly what an ideal metric would look like, i.e. uncorrelated with number of features so as not to be biased by methods that select fewer or more features, has a dynamic range that allows for informative selection of methods, etc. This is implied by reading through the excluded metrics in table 1, but if stated up-front then it would be clearer.

Expressing the standard deviation as the size of the square across figures makes it very hard to discern just how variable the correlations are. More so, because the size of the square is inversely proportional to the variation. This is likewise confounded with the colour range in Figure 1B and Figure 3B/C, in that white is in the middle of the range (correlations ~0), and so has a poor contrast with the light coloured background of the figure panel. I would suggest a supplementary figure that displays the variance of the correlations as a single value - this would significantly aid clarity of presentation. This is especially problematic because more variable methods actually have smaller points, which is counter-intuitive. I understand that the authors might want to emphasis the methods with a low correlation and low variance, but this current visualisation doesn't quite hit the mark.

At the end of the penultimate paragraph of the section "The number of selected features affects performance", the authors state that "...this trend holds across all data sets, with more features required to achieve high scores...". This suggests that cell type diversity is roughly proportional to the number of features required to get a good integration and query-reference mapping. Could the authors discuss briefly at what point this could saturate, if at all? (modulo including all genes).

The text regarding the overall results is a little confusing (page 10). This is because the ordering in Figure 4A is not by highest overall ranking, but by the average overall score. So is Wilcoxon or Seurat-VST the highest ranked method in this benchmark? At the moment the text and figure are not 100% harmonious. Perhaps this discrepancy comes from the 'Mean ranks' panel of Fig4A to declare the "winning method" rather than the "Overall score" which was used to order the methods in the figure. This requires some harmonisation to be 100% transparent and clear.

Page 34 - the sentence "...feature sets (have an insufficient dynamic range), were overly correlated with the number of features..." - could the authors define concretely what value or range of values they considered "overly correlated"?

Reviewer #2 (Remarks on code availability):

The code is extensive, and appears well documented. The inclusion of a wiki page walks potential users through the installation steps for third-party tools (conda, nextflow), and explains how to run the pipeline using Nextflow. There is also an explanation of the API for re-running the benchmarking, however a concrete example of key steps in the pipeline would aid clarity.

I have not attempted to run the code, due to the scale of the compute requirements.

Reviewer #3 (Remarks to the Author):

I am satisfied that the authors completed the experimental plan as described, exploratory analyses are appropriately described and caveated and the conclusions are justified. The manuscript and some figures have various typos/misspellings/missing words and would benefit from a careful edit.

Reviewer #3 (Remarks on code availability):

The code is available and documented.

Reviewer #4 (Remarks to the Author):

Although this version has improved a lot, I still have the following minor comments:

1. Page 2/41: Reference 2 is from 2018, and the authors cite the integration of 250 tools (the previous version mentioned 200), which seems somewhat inappropriate.
2. Page 3/41: "avoid include several metrics which provide the same information" should be replaced with 'avoid including multiple metrics that provide the same information.'
3. Page 7/41: "averaged over five sets" does not clearly explain what it represents. Additionally, although the authors mentioned four methods, they did not explain how the normalization was performed. The authors should use strict mathematical formulas to express this.
4. Page 24/41: 'less than 10 percent of cells within that label, expressed in more than 80 percent of cells outside the label or with a p-value above 0.1. Next, the remaining features are sorted by estimated log-fold change, and the top 200 features are selected. The final feature set is the intersection of the features selected for each label.' Are the thresholds set too arbitrarily, such as $p > 0.1$ and the top 200?" Do different parameter settings affect the results

Version 4:

Decision Letter:

8th Feb 2025

Dear Dr Theis,

I am pleased to inform you that your Registered Reports, "Feature selection methods affect the performance of scRNA-seq data integration and querying", has now been accepted for publication in Nature Methods. The received and accepted dates will be 7th Jun 2023 and 8th Feb 2025. This note is intended to let you know what to expect from us over the next month or so, and to let you know where to address any further questions.

Over the next few weeks, your paper will be copyedited to ensure that it conforms to Nature Methods style. Once your paper is typeset, you will receive an email with a link to choose the appropriate publishing options for your paper and our Author Services team will be in touch regarding any additional information that may be required. It is extremely important that you let us know now whether you will be difficult to contact over the next month. If this is the case, we ask that you send us the contact information (email, phone and fax) of someone who will be able to check the proofs and deal with any last-minute problems.

You may wish to make your media relations office aware of your accepted publication, in case they consider it appropriate to organize some internal or external publicity. Once your paper has been scheduled you will receive an email confirming the publication details. This is normally 3-4 working days in advance of publication. If you need additional notice of the date and time of publication, please let the production team know when you receive the proof of your article to ensure there is sufficient

time to coordinate. Further information on our embargo policies can be found here:
<https://www.nature.com/authors/policies/embargo.html>

If you are active on Twitter/X, please e-mail me your and your coauthors' handles so that we may tag you when the paper is published.

Best regards,
Lei

Lei Tang, Ph.D.
Senior Editor
Nature Methods

** Visit the Springer Nature Editorial and Publishing website at http://editorial-jobs.springernature.com?utm_source=ejP_NMeth_email&utm_medium=ejP_NMeth_email&utm_campaign=ejp_Nmeth for more information about our career opportunities. If you have any questions please click [here](mailto:editorial.publishing.jobs@springernature.com).

Open Access This Peer Review File is licensed under a Creative Commons Attribution 4.0 International License, which permits use, sharing, adaptation, distribution and reproduction in any medium or format, as long as you give appropriate credit to the original author(s) and the source, provide a link to the Creative Commons license, and indicate if changes were made. In cases where reviewers are anonymous, credit should be given to 'Anonymous Referee' and the source. The images or other third party material in this Peer Review File are included in the article's Creative Commons license, unless indicated otherwise in a credit line to the material. If material is not included in the article's Creative Commons license and your intended use is not permitted by statutory regulation or exceeds the permitted use, you will need to obtain permission directly from the copyright holder.

Editor comments

I have discussed the reviewer reports in detail with our chief editor and other members of the Nature Methods team, with a view to 1) identifying key priorities that should be addressed in revision, and 2) overruling reviewer requests that we think fall beyond the scope of the study. Please see the attached file which contains the reviewer reports annotated with our editorial recommendations. We invite you to revise your Stage 1 proposal to address these concerns (before you embark on the proposed experiments).

Thank you for your additional guidance on the reviewer comments. We have replied to the reviewers below but also address your specific points here.

Reviewer 1

We think that the referee raises important concerns regarding the comprehensiveness of this study, however we are unsure about the efforts needed to fully address the concern about the effects of the biological factors on the performance of feature selection methods.

If this takes substantial efforts or you believe are technically impossible or unlikely to yield a meaningful outcome, please provide reasonable arguments.

We have addressed the reviewer's comments about considering the effects of biological factors on feature selection and integration by including another subset of the Human Lung Cell Atlas as a dataset in the study. This will allow us to expand the planned analysis of how a more constrained biological setting affects performance and the differences in selected features when methods are applied to a subset rather than the full dataset. Our planned analysis is now more clearly explained in the responses and in the text.

Reviewer 2

We agree with the referee that the inclusion of synthetic data in the comparison would strength this work and make it more complete.

Yes, we agree. To address this point, we have included a simulated dataset from the splat simulation model and will consider how methods and metrics perform differently on this dataset compared to the real data.

We don't think it is necessary to include alternative chemistries in this comparison study.

Thank you for your clarification on this point. We have replied to the reviewer with an explanation of why we believe this to be out of scope.

Reviewer 3

We think it will be important to include more recently developed methods in the comparison study, as also suggested by reviewer 4.

We have addressed the reviewer's comments about focusing too closely on deep learning-based integration methods by adding integration using Harmony followed by query mapping using Symphony. This represents a different class of integration approaches and will allow us to see if feature selection methods perform similarly for this type of integration. These changes also address the similar comment from Reviewer 4.

Reviewer 4

There is one suggestion we find out of scope, and we do not expect the authors to use datasets from tumor TME to evaluate the performances of feature selection methods.

Thank you for your clear guidance on this point. We have replied to the reviewer that we feel this would be out of the scope of the study.

Reviewer 1

I have reviewed the proposed benchmarking paper according to the criteria provided by the editor.

1) The importance of the comparative analysis and relevance for a broad, multidisciplinary audience.

The ultimate goal of this manuscript is to improve single cell data integration and querying, two important tasks in the broader field of single cell data analysis. Given the importance of single cell technologies in biology and medicine, the notion of improving data integration and querying is clearly significant.

Thank you for your comments and agreement with the importance of the study.

2) The rationale, plausibility and comprehensiveness of the proposed comparative analysis.

This manuscript specifically proposes to benchmark feature selection methods in the context of single cell data integration and querying. Feature selection is a critical step in data preprocessing for single cell data analysis, as **all** downstream analysis tasks (visualization, clustering, integration, querying, etc.) operated on the selected feature set. This study therefore has strong rationale.

Thank you for agreeing on the importance of feature selection and supporting the rationale for the study.

The proposed study as is, is also plausible -- the senior authors have executed a very similar benchmarking study on integration methods previously, and are building on their previously successful benchmarking work.

Thanks for your belief in the plan for our study and recognising our prior work.

I have strong issues with the comprehensiveness of this study as designed, however. Central to the problems of integration and querying is the notion that cell populations/types have varying levels of heterogeneity driven by both biological factors (e.g. conditions) and technical factors (batch, etc.), as pointed out by the authors. Feature selection plays an important role in integration in part because it defines the input features which are used both to find matching cell types across batches, and whose variation it is assumed to capture the biological variability within a population. One can therefore easily imagine that the "optimal" set of input features is a function of cell types present, organ systems, technology, etc. A large blindspot for the current benchmark proposal is studying what the effect of these important biological factors (cell type, organ, technology, species) is on both the 'performance' of the feature selection methods, and whether certain methods are better suited to capture features for certain cell types/systems/etc. than others.

We agree that the performance of feature selection methods may be related to the characteristics of the datasets they are applied to. When selecting datasets for the study, we tried to cover a range of tissues and scenarios from a small integration of a few batches to tissue-scale atlas efforts. Once the full results from the benchmark are available, we will connect the performance of methods to technical factors (such as number of cells or batches) as well as biological factors (such as the number or diversity of cell types). In particular, we will consider the relative performance of methods on the full HLCA dataset compared to specific compartments. This will provide insights into the performance of feature selection methods when biological diversity is constrained as well as the effectiveness of lineage-specific integration. To extend this comparison, we have included the epithelial compartment of the HLCA as a dataset as well as the immune compartment we included previously. We believe that comparing results between datasets as well as between subsets of the HLCA will allow us to answer the important questions you raise here. We have added this as a focus of analysis in the text.

Another focus of analysis will be to compare between the full HLCA dataset and subsets representing the immune and epithelial compartments. This will allow use to see how feature selection methods perform differently when a dataset is limited to more similar cell types.

The full benchmarking pipeline, including scripts implementing all the of the metrics, will be publicly available so that users can apply them to their own data if they have specific questions of interest.

As the senior authors are obviously aware given their work on HLCA and the HCA in general, there are huge atlases from many different species that are generated from multiple groups, technologies, etc., that could be leveraged to study the effect of key biological factors like cell type on feature selection. That would, in my mind, increase the value of the benchmark paper significantly. While the authors propose to use 8 different datasets for the benchmarking, it would be much more informative if the authors also chose bigger datasets like from the HCA or Tabula muris/sapiens, where they could systematically vary biological factors like cell type (without jumping across different datasets to do so). Such an analysis would then allow the authors to provide more generalizable recommendations for how to select the best feature selection approach for their own work.

We feel that the selected datasets already allow us to answer the biological questions raised (as described above) and include large, well-annotated atlases such as the HLCA (part of the HCA). While the Tabula Muris/Sapiens are valuable resources, they are of comparable size to datasets already included in the study (Tabula Sapiens has around 480,000 cells, slightly less than the number in the HLCA (560,000 cells) and Reed breast (530,000 cells) datasets, and Tabula Muris has around 350,000, fewer than several of the current datasets). These numbers would be further reduced if we were to subset to specific tissues. Additionally, because they are constructed as a single study, they are likely to have lower batch complexity than some of the current atlas datasets which are made up of several public datasets with different protocols etc. We also

include datasets such as the Reed breast data which come from a single study but have significant internal variation. We expect that technical (technologies, batches) and biological (number of cells, cell types) factors covered by the existing datasets to have more of an effect than differences between tissues from one study. Adding a dataset also involves a significant computational cost, resulting in approximately 100 additional integration runs. For these reasons, we don't believe adding these datasets would add to our study.

I am also *very* surprised the manuscript doesn't propose to vary the number of features selected (given the same feature selection approach) -- as an end user, it is completely unclear how the number of features selected is chosen (particularly for methods where that choice must be made explicitly by the user). It is extremely odd for the proposed benchmark to actively decide not to vary this for each chosen method, given it is a benchmarking paper about feature selection. This needs to be rectified by the authors to be a valuable study to the community.

Thank you for raising this important point. Although the main focus of the study is comparing between methods, we had also intended to investigate the effects of the number of selected features for a subset of methods and have expanded on this section based on your comment. We propose to limit this part of the analysis to only some methods in order to limit the computation required for what is already a computationally intensive study. The revised study already considers over 900 integration runs and around 30,000 metric scores, considering the number of features for a method would require 5-10 times the number of integrations, depending on how many points are chosen. We expect the effect of the feature set size to be relatively consistent between methods, particularly in the typically used range of 1000 to 6000 features [1]. We have added a section to the text about this part of the analysis.

The majority of feature selection methods evaluated here require the user to specify the number of features to select. But, while this can clearly have an effect on downstream analyses, there is no clear guidance as to how many features should be selected and how that is related to biological factors such as the diversity of cell types in the dataset. To address this question we will include different numbers of features for a subset of commonly used methods in our evaluation. This will allow us to see how performance changes with the number of features. While it would be interesting to have this data for all methods, each additional number of features is a significant computational cost. We chose to limit this part of the analysis by methods rather than dataset as it will allow us to see the effect of number of features across datasets while we expect the trend in number of features to be similar for different methods. The results of this analysis will inform the number of features that we use for all methods in the full evaluation.

3) The soundness and feasibility of the methodology and data analysis pipeline (including statistical power analysis where appropriate).

Given the caveats mentioned above regarding lack of comprehensive exploration of the dependence of results on cell types / organ systems / etc. mentioned above, the proposed evaluation metrics, reuseable workflows, etc., are sound and feasible, particularly given the senior authors' past work on benchmarking single cell integration methods.

Thank you for your belief in our analysis pipeline.

4) Whether the clarity and degree of methodological detail would be sufficient to replicate exactly the proposed experimental procedures and data analysis pipeline.

I have no concerns on replicability of the work - the senior authors have demonstrated from their past work that they generate reproducible benchmarking studies.

Thank you for recognising our commitment to performing a reproducible benchmark.

5) Whether the proposed comparative analysis is subject to any bias.

None that I can see.

Reviewer 2

Zappia et al. present a plan to perform a benchmarking analysis of feature selection for scRNA-seq, and assess the impact on data integration and reference-query mapping. They provide a comprehensive list of potential metrics, and importantly, include plans for a comparative analysis of such metrics to select the “most relevant”. Broadly, the plan is thorough and has the potential to greatly inform both standard practices in single-cell atlas building, query-reference mapping, as a reference for newcomers to scRNA-seq analysis to select an appropriate feature selection algorithm.

Thank you for your understanding and appreciation of the study

That said, there is room for clarification, and some adjustments to their current strategy are recommended.

Many of the evaluation metrics require a ground truth, which the authors here define as the reference data in the reference-query problem setting. However, this approach assumes the reference is unbiased (vs. strong assumption), and so opens the door to the metrics also being biased. The benchmarking strategy would greatly benefit from simulations and/or synthetic data to define ground truths; the latter are suggested due to the general limitations on realism for simulated single-cell gene expression data on the scale and complexity of atlas data sets. The inclusion of simulations, then comparing the simulation vs. real data vs. synthetic data sets will provide a good indicator of the consistency of the benchmarking metrics. If they are inconsistent, then it shows that either the simulations were unrealistic (need better simulations) or that the metrics or feature selection methods are not consistent.

Thanks for the suggestion! We agree that using the annotations provided by the original authors as our main source of truth information may introduce bias into the study. While simulations provide a definite ground truth, they can not reproduce all aspects of real datasets and we have seen in previous studies that often provide less of a challenge for methods [2]. Despite the limitations, we agree that adding simulated datasets would add to the study. We propose adding a dataset from the splat simulation model [3] which can flexibly simulate a range of scenarios and which we have previously used for integration benchmarking. A section describing the splat simulation has been added to the manuscript:

Evaluations using real datasets provide the most accurate assessments of performance but they also present challenges as they rely on ground truth from previous analyses which may be incomplete or biased towards the methods that were originally used. Some of these concerns can be addressed by simulations where a definite ground truth is known. We have created a simulated dataset using a modified version of the splat simulation in the Splatter package. This has been designed to represent a scenario where a tissue is measured using three different technologies (two batches each) in two conditions. These “technologies” measure a medium number of cells at medium depth (Batch1, Batch2), low number of cells at high depth

(Batch3, Batch4) and high number of cells at low depth (Batch5, Batch6), with the first two comprising the reference and the last one the query. The simulation contains 10 cell labels including a progenitor differentiating along two trajectories (one with an intermediate cell type only present in the query) and six discrete cell types that differ in number of cells, number of differentially expressed genes and number of detected features. The discrete groups include a rare population and perturbed state both only present in the reference. To increase the variability in the simulation we added additional label-specific noise factors to the model which are applied just prior to generating counts. After preparation, the splat dataset contains 9984 features, 30041 reference cells (4 batches, 7 reference labels) and 69936 query cells (2 batches). The “Intermediate”, “Rare” and “Perturbed” labels are only present in the query.

Is the expectation that a random set of features would be less performant than the feature selection methods a reasonable one? Random subsets of genes might perform surprisingly well as they will capture a lot of structure based on gene-gene correlations. Particularly as the random subsets would need to be matched to selected features on composition, such as gene expression level distributions, sequence-ability, etc, to be a fair null comparison. When comparing to random subsets, how many permutations of the random feature set are used? This doesn't seem to be defined and is implicitly = 1. This could be limited, and prone to sampling variation. Moreover, does a random subset need to be computed for each method to provide a valid null comparison?

We have included randomly selected features to replicate a worst-case, naive method (i.e. one that does not use any relevant information from the dataset), rather than a “null” feature set. They are primarily used to help calculate baselines for comparing the real methods, rather than being of interest themselves. We expect them to contain some information because (as you correctly point out) genes are highly correlated and informative genes will be selected by chance, especially for larger random sets. As a better “null” set we include features selected by the scSEIndex method which aims to select genes that are stably expressed across a dataset. We also include other naive methods such as selecting the most highly expressed genes. To control for the variability in random selection, we consider 5 random feature sets and have added a sentence to the text describing this (*“To control for variability in selecting random features we always include five random feature sets selected with different seeds”*). Because random sets are not directly compared to specific methods there would be no gain in matching them.

It is not clear why different implementations of the same method would be required. Unless of course these are not implemented the same. In which case, they should not be labelled as the same method and divergence between algorithms should be clearly stated for all users, especially newcomers to the field. The justification for their inclusion based on pre- processing differences seems rather weak, as the focus is then on the pre-processing, and is not related to the feature selection algorithms/methods per se.

While different implementations of a feature selection method should produce the same feature sets, it is likely that this is not the case and testing this is part of the motivation for including different

implementations of a few, commonly used methods. We consider a “method” to include pre-processing steps (starting from a raw count matrix) as described in associated documentation for two reasons: 1) Pre-processing is required and varies between methods so there is no one standard pre-processing that would be suitable for all methods, and 2) the suggested pre-processing represents both typical and (presumably) best practice usage. We aim to benchmark how typical usage of these methods and packages affects integration and query mapping, not the interaction between pre-processing and feature selection which (while potentially interesting) would require a separate study. We believe that providing results for different implementations of common methods will clarify these issues and provide insights for readers. These points have been mentioned in the text:

Any pre-processing steps required prior to feature selection are considered as part of the method. We use the steps suggested in the documentation for each method as those are recommended by the authors and represent the most likely real-world usage.

The focus is largely on data sets generated by drop-based scRNA-seq chemistries. While this may be the chemistry du jour, inclusion of legacy and alternative chemistries warrants inclusion. Especially as future atlas/meta- analyses will make use of publicly available data.

We have not deliberately focused on this technology but rather selected publicly available, curated datasets. In some cases, these datasets contain multiple technologies and protocols, specifically the pancreas dataset. Other datasets that have a wider variety of technologies may not be suitable for other reasons. We do not see comparing between technologies as a focus of the study as droplet-based technologies are by far the most commonly used currently and we cannot predict what technologies may be used in the future. The computational load of this study is already high and there are other studies that look more closely at differences between protocols. If we do notice any differences in feature selection performance for different technologies, we will comment on that in the discussion of the final study.

For several non-excessive variability methods, an initial feature set is required. Across these methods, inconsistencies arise in the initial feature selection method, which could be a huge source of (a) variation and (b) bias. Instead of the current categorisation, methods could be split into primary and secondary methods, with the secondary methods are applied to the out from the primaries to give a fairer comparison. The natural cost is many more comparisons, but this strikes me as being considerably fairer and less prone to bias and inconsistencies.

As described above we consider a “method” to be a combination of the feature selection and the pre-processing recommended by the method/package authors, including any initial feature selection. We believe this best represents real-world usage and provides a better estimate of the performance of methods when used by the typical analyst. Considering all combinations of initial and secondary methods would greatly increase the computational requirements (approximately 30 method combinations leading to

hundreds more integrations) for what is already a computationally intense study and be inconsistent with how we treat other pre-processing steps. For those methods that do require an initial feature set, we will explore the overlap with the final set to determine what effect this may have.

The supervised methods rely on published methods, but omits biologically motivated feature sets. For instance, all transcription factors, or all cell surface expressed genes, as these may also be highly characteristic of the cell type diversity in an atlas/sample.

The focus of the study is on unsupervised methods as we believe they represent typical usage, both because of common practice and because supervised methods often require cell type labels that may not be available until after integration. The one supervised method we include is marker genes based on the Wilcoxon rank sum test. We appreciate your suggestion of an alternative feature set based on prior biological knowledge and we have addressed it by including features from a database of transcription factors as a method. This is described in the text:

In addition, we include a supervised method based on known transcription factors. We downloaded a database of human transcription factors from The Human Transcription Factors website (<http://humantfs.ccb.utoronto.ca/index.php>) and selected 1639 genes where the “Is TF?” field was equal to “Yes”. The intersection of this list with the genes in each dataset is then used for evaluation.

For each evaluation metric, a short explanation of whether a low or high score is better/worse would greatly benefit readers not familiar with the technicalities of each evaluation method.

All metric scores are adjusted so that higher scores are better with raw scores between 0 and 1. For final comparisons, each metric is rescaled based on the baseline method with 0 representing the worst baseline and 1 the best baseline. Other methods may exceed this range if they perform better/worse than the baselines. This is now more clearly stated in the text:

All metrics are designed so that a raw score of 0 represents the worst possible performance and a raw score of 1 the best possible performance.

A more principled selection of metrics would be most informative for readers and for the research team in terms of interpreting results and making recommendations. This indicates a pilot experiment to select informative and non-redundant metrics is required before undertaking the full benchmarking. While the authors propose to perform a ‘metric selection analysis’ there is no details on how this will be achieved and is one of the biggest omissions of the study design. If anything, this metric selection process should have already been performed.

We agree that not all metrics may be informative and/or may measure the same information, for this reason, we proposed the metric selection step of the study. We chose to include this as part of the main study rather than a separate pilot as we believe the results will be of interest to readers and it will encourage the use of a similar process in other benchmarks. As described in the text, the metric selection will be performed using the scores of random feature sets of different sizes on the selected datasets. Random feature sets will be used to avoid biasing the feature selection to any real method. This process will allow us to investigate factors such as the sensitivity and range of metrics, correlations between them and associations with technical aspects such as the number of genes, cells or labels. By considering the results of this analysis, we will select a final set of metrics to use that are informative and non-redundant. We have added more detail to the metric selection section to help clarify this step.

Metrics will be removed if they can not distinguish between feature sets (have an insufficient dynamic range), are overly correlated with the number of features, cells or cell types, or show other unfavourable characteristics. Following this step, if there are still highly-correlated metrics we will select a subset to use. The outcome of the metrics selection process will be a non-redundant set of features with fewer biases covering the categories of interest for the final benchmark.

An explanation of the assumptions of each method and evaluation metric would greatly benefit the interpretation for the reader. More concrete details are required for many metrics, for instance, qualitative thresholds such as “fewer” are ill-defined and should be made explicit.

The methods and metrics we use here have been previously published, and in most cases, we use an existing implementation. For that reason, we only provide brief summaries of each here and refer to the original publications for more detail. We feel that providing extensive details on each of these would be beyond the scope of this study. We have read through the methods sections to add further detail where it was lacking.

For several evaluation metrics, it is not clear what the random distribution of values would look like. While the pilot data attempts to show this I am not convinced the “null” methods used are truly “null” in the sense that the feature sets will be depleted for “true” cell identity/state/etc defining gene sets.

We agree that it is often difficult to establish the real ranges of evaluation metrics. This is part of the motivation for both the metric selection process (to see how they perform over a wide range of inputs) as well as the scaling procedure based on selected baseline methods. Selecting a set of uninformative features is indeed difficult due to the correlation between genes and the fact they all perform some biological function. To address this we include the scSEIndex method as a baseline which aims to identify features that are stably expressed across the dataset.

Presentation of analyses: The central tendency and the spread of values should be reported. e.g. 2 methods may have moderate scores across metrics, but 1 with a smaller spread than the other. The smaller spread would imply greater consistency across measures, and thus more confidence in its ranking compared to the method with a large spread.

Thank you for the suggestion, the spread of results is indeed important. We will make sure this is displayed in some way and consider how it can be included in the ranking of methods. For the final evaluation, we will include variation between metrics from the same category as well as across datasets.

Minor/method/metric-specific comments

The authors refer to “flavours” in the Seurat and cellranger excessive variability methods. Is this a typographical error (function?) or the improper use of a colloquialism?

The “flavour” term comes from the name of the function argument in the Scanpy package that controls which approach is used. We agree this is unclear for people not familiar with the package and have replaced it with “method”.

For the NBumi method the authors state they will select features with $p < 0.01$ or the 500 lowest p-values, whichever gives the larger feature set. This could be a source of bias, and should be justified with e.g., a sensitivity analysis, especially as most of the metrics/assessment is performed down-stream of feature selection, i.e., in terms of reference- query mapping.

This step was included in the implementation of the NBumi method to prevent an error on the unrealistically small test datasets used during development of the pipeline. We found that due to the very small number of cells in these datasets no features had p-values less than 0.01 and no genes were selected. To prevent this, we added a check and the selection of 500 features. We don't expect this check to be necessary on the real test datasets which have much larger numbers of cells.

How does the scanpy implementation of the Pearson residuals method differ from that of the original by the Irizarry group? Why is the original not included?

The Scanpy implementation of the Pearson residuals method was contributed to the package by the authors of “Analytic Pearson residuals for normalization of single-cell RNA-seq UMI data” [4]. We also include the “scry” package which implements the deviance residuals method in “Feature selection and dimension reduction for single-cell RNA-Seq based on a multinomial model” [5] (which we believe is the paper you are referring to) and was created by many of the same authors.

For the singleCellHaystack method the authors state that they will use the Seurat HVG method for the initial round of PCA. Does this just return a feature set that is strongly enriched for the original set? Is this

tautological? Should this use each of the different feature selection methods for the initial feature selection step to ensure that it doesn't just recapitulate the rankings of all other methods?

We feel this is similar to one of the points from reviewer 1 so have included a similar response here:

We consider a “method” to be a combination of the feature selection and the pre-processing recommended by the method/package authors, including any initial feature selection. We believe this best represents real-world usage and provides a better estimate of the performance of methods when used by the typical analyst. Considering all combinations of initial and secondary methods would greatly increase the computational requirements for what is already a computationally intense study and be inconsistent with how we treat other pre-processing steps.

In addition, comparing the overlap in sets from different methods is one of the analysis steps we intend to perform and should show how closely these secondary methods are related to the initial feature selection method. This is now described in the analysis section:

Further analysis will look at the similarity between methods by considering the overlap in selected sets. This will help to explain method performance as we can relate differences in selected features to differences in performance and inspect which features top-performing methods select that others don't or if rather than selecting better features they exclude more uninformative features. We will also use this analysis to check the similarity feature sets from “secondary” methods such as SingleCellHaystack that operate on an embedding produced by an early feature selection to the initial feature set.

For the Hotspot method, as with singleCellHaystack, this has the potential for a circular result. The same applies to triku – what feature set is used for the initial PCA?

For both HotSpot and triku the PCA space is constructed based on all available features as that is what was suggested by the documentation. We have updated the text for these methods to clarify this detail.

For the Batch PCR evaluation are the regression coefficients standardised? Are the components matched, i.e. does PC1 correspond to integrated dimension 1? Or is this the sum over all coefficient-weighted variances? This will show that variance is removed and from which components. Should the test be on a component-wise basis to demonstrate that non- batch axes of variance are preserved? More details required here.

The batch PCR metric is a sum over all coefficient-weighted variances to measure the overall removal of variance that is linearly correlated with the batch covariate. The following text from Leucken et al. [2] (whose implementation we use) describes this metric in more detail:

“Principal component regression, derived from PCA, has previously been used to quantify batch removal. Briefly, the R^2 was calculated from a linear regression of the covariate of interest (for example, the batch variable B) onto each principal component. The variance contribution of the batch effect per principal component was then calculated as the product of the variance explained by the i th principal component (PC) and the corresponding $R^2(PC_i|B)$. The sum across all variance contributions by the batch effects in all principal components gives the total variance explained by the batch variable as follows:

$$\text{Var}(C|B) = \sum_{i=1}^G \text{Var}(C|PC_i) \times R^2(PC_i|B),$$

where $\text{Var}(C|PC_i)$ is the variance of the data matrix C explained by the i th principal component.”

For iLISI how sensitive is this the proposed approach to the number of original batches? What is the null distribution, a multinomial (binomial for 2 batches?)

A raw LISI score measures the number of neighbours of a cell until the same label is observed twice and can range from 1 to the number of labels. We use the graph LISI implementation from Leucken et al. [2] which rescales this range to between 0 and 1 as described here:

“As LISI scores range from 1 to B (where B denotes the number of batches), indicating perfect separation and perfect mixing, respectively, we rescaled them to the range 0 to 1. For iLISI and cLISI this involved a two-step process. First, we computed the median across neighborhoods per method:

$$\text{cLISI} = \text{median } f(x), x \in X; \text{iLISI} = \text{median } g(x), x \in X.$$

Second, we rescaled the LISI scores as follows:

$$\text{cLISI} : f(x) = \frac{B-x}{B-1}$$

where a 0 value corresponds to low cell-type separation and

$$\text{iLISI} : g(x) = \frac{x-1}{B-1},$$

where a 0 value corresponds to low batch integration.”

Neither Leucken et al. who implemented the scaled, graph LISI we use here or Korsunsky et al. [6] who originally proposed the LISI metric for batch integration describe the sensitivity of the metric to the number of batches (although Korsunsky et al. do explore the effect of uneven batch sizes). This is something that will be checked as part of our metric selection stage.

For the CMS tests, does the implementation assume a specific distribution, or is it undefined as in the K-sample Anderson-Darling test?

- Refer to original paper
- Pretty sure it uses Anderson-Darling

The CMS metric uses the Anderson-Darling test, from the paper by Lütge et al. [7]:

“For each cell, batch-wise distance distributions to the cell’s knn are retrieved. cms scores the null hypothesis that the distances originate from the same distribution (across batches) using the Anderson–Darling test.”

The statement that “The scanpy methods is ...an example of the currently accepted best practise...” – I would like to see a citation or evidence to support this statement.

We have adjusted the language here and added citations. It now reads:

The scanpy method is included as an example of current standard practice...

Reviewer 3

The proposed study aims to evaluate the effects of feature selection on single-cell RNA-seq integration (batch correction) and query mapping with label transfer. Overall, the study seems carefully designed and comprehensive and the findings would be useful to the single-cell community.

Thank you for your kind words and understanding of the potential of the study

My main question/concern has to do with the limited choice of integration method used for testing (scVI and its extension scANVI), I believe this analysis would be much more powerful if it considered a handful (even just 1-2) of other commonly used integration methods – a recent benchmarking of integration methods coming from the same group, for example (manuscript ref 3) found Scanorama and fastMNN to be among the top performing methods. Anecdotely, variants of MNN and Harmony among others are still being commonly used in single-cell analyses. My suggestion would be to include another integration method in this analysis to widen the relevance and applicability.

Firstly, we would like to emphasise that the aim of the study is to compare feature selection methods, not to evaluate integration methods, which has been done previously [2,8,9]. This, as well as reducing the amount of computation required, provides the motivation for wanting to limit the number of integration methods used. There are also other limitations to consider, including that it must be possible to map query samples (which relatively few integration methods support) and scale to large datasets (which has been shown to be difficult for some methods including Seurat). For these reasons, we selected the deep-learning-based methods in the scvi-tools package which have been used for several atlas-building efforts and have been shown to perform well, particularly for large integration tasks. These methods also allow us to compare between unsupervised (scVI) and semi-supervised (scANVI) variants and accept raw counts as input, which removes the need for pre-processing. However, we accept that this is a small subset of integration approaches and there is likely to be an interaction between features selection methods and the integration used. To address your comment, we have added integration with Harmony [6] followed by query mapping with Symphony [10]. This represents an alternative class of integration models based on a corrected PCA space that should show the generalisability (or not) of feature selection performance. This additional integration method is described in the text:

As an example of alternative approaches based on correcting a PCA space, we include integration with the Harmony followed by query mapping using the associated Symphony approach. This represents an alternative class of integration methods and lets us see if the performance of feature selection methods is consistent when compared to deep learning-based integration. As suggested in the documentation, we provide Harmony with normalised expression values rather than raw counts. Counts are first normalised to counts per 10,000, then log-transformed. The dataset is then subset to the provided features and scaled with a maximum value of 10 (per feature) before calculating 30 principal components that are provided as

input to Harmony. For query mapping using Symphony, log-transformed normalised query data is provided (scaling is performed as part of the mapping function). Data pre-processing steps are performed using functions in the scanpy package, and integration and query mapping are performed using the symphony package (<https://github.com/potulabe/symphony>).

Additionally, I have a few minor notes:

- For the method singleCellHaystack, why do you plan to use Seurat defined HVGs for the PCA? Why not run PCA with all features so as to not to bias the chosen features?

We repeat a similar response to the similar comments from reviewers 1 and 2.

We consider a “method” to be a combination of the feature selection and the pre-processing recommended by the method/package authors, including any initial feature selection. We believe this best represents real-world usage and provides a better estimate of the performance of methods when used by the typical analyst. Always using all features for PCA would simplify this issue but still means a departure from the recommended usage for SingleCellHaystack.

- It would be useful for the authors to note why they are choosing each of the query sets. In some cases it seems obvious (when there are only two major batch types for example), but in others the choice seems arbitrary – would you expect any difference in results if another outgroup was chosen?

You are correct that the choice of query batches and unseen cell labels could affect the results. The most complete evaluation would be to check every combination of batch and cell label, but that would make an already large study computationally infeasible, and we feel it is more important to cover a range of scenarios and datasets. As you note, for some datasets, choosing the query set was relatively straightforward. For example, the Reed breast dataset has donors with different BRCA status, and the NeurIPs dataset includes different sites. When this was not so clear we tried to select batches with similar characteristics as the query. Labels to be only present in the query were chosen by first looking for cell types that were enriched in the chosen query batches compared to the reference. When these did not exist, or to extend those we found, we tried to choose labels that would present a range of challenges. For example, holding out a rare cell type, part of a lineage or a perturbed state. More detail about this has been added to the text:

Query batches were chosen by selecting a set of batches with shared characteristics different to the remaining reference samples such as technology, timepoint or location. The unseen populations only present in the query were chosen by first looking for labels that were enriched in the query batches and then selecting labels that would present different challenges such as rare or perturbed cells.

Reviewer 4

Luke Zappia et al. benchmarked gene selection methods for integration and reference usage on single-cell data. Although this study evaluated various datasets with different feature selection methods/algorithms and various metrics (integration-batch effect, integration-biological conservation, mapping quality, label transfer, and unseen population prediction), I have several questions and concerns about this manuscript, which I think are important to be addressed.

Thank you for taking the time to review the study and make comments

Major comments

1. The title of the study said “Feature selection methods affect the performance of scRNA-seq data integration and querying”. However, the authors only focus on deep learning integration methods. I think the author should include other popular integration methods, such as scGen, Harmony, Seurat, fastMNN, Scanorama, and BBKNN. All these methods performed well in one of your previous articles (Malte D. Luecken et al, 2022, Nature Methods).

This point was also raised by reviewer 3, we have included the same response here (without the text snippet):

Firstly, we would like to emphasise that the aim of the study is to compare feature selection methods, not to evaluate integration methods. This provides some motivation for wanting to limit the number of integration methods used, as well as reducing the amount of computation. There are also other limitations to consider, including that it must be possible to map query samples (which relatively few integration methods support) and scale to large datasets (which has been shown to be difficult for some methods including Seurat). For these reasons, we selected the deep-learning-based methods in the scvi-tools package which have been used for several atlas-building efforts. These methods also allow us to compare between unsupervised (scVI) and semi-supervised (scANVI) variants and accept raw counts as input, which removes the need for pre-processing. However, we accept that this is a small subset of integration approaches and there is likely to be an interaction between features selection methods and the integration used. To address your comment, we have added integration with Harmony [6] followed by query mapping with Symphony [10]. This represents an alternative class of integration models based on a corrected PCA space that should show the generalisability (or not) of feature selection performance.

2. Feature selection is indeed a fundamental step in single-cell RNA sequencing (scRNA-seq) data analysis. However, the significance of comparative analysis on this topic has been somewhat diminished by the publication of (Malte D. Luecken et al, 2022, Nature Methods)

We disagree that Luecken et al. fully addressed the question of feature selection for scRNA-seq integration. While that study did compare selected features to all features, the focus was on evaluating the integration methods themselves and only a single general-purpose feature selection was used. This study extends the finding that feature selection can affect integration by comparing several feature selection methods, including other statistical approaches, as well as some that take advantage of other features of scRNA-seq data. As a second original contribution, we also consider the effect of feature selection on downstream usage (mapping, label transfer and unseen population detection) of the integrated atlas which was not part of the Luecken et al. study. With the upcoming large sets of integrated atlases and usage thereof, we feel this can strongly help improve on method choice and resulting atlas quality. These points are covered in the introduction of the revised manuscript.

3. In real data analysis, we will have different focuses for different tasks. When evaluating different feature selection methods, the author should provide recommendations on which HVG selection methods are best for different tasks instead of simply giving an overall score.

Good point. Here, we focus on the effect of feature selection on integration to construct an atlas and its downstream use through query mapping. We divide the evaluation metrics into five categories (batch correction, biological conservation, mapping quality, label transfer and unseen population detection). Some of these metrics do touch on downstream tasks such as clustering or detecting new populations. We will provide summary scores for each category so that readers can interpret the results based on their particular use case. Feature selection specifically for other tasks (such as clustering) has already been considered in previous studies.

4. Although the author used various feature selection methods, it is important to know how many genes are selected in common or differently with different methods. The author should provide a detailed table or figure for each dataset to illustrate this.

Thank you for the comment. This is something we had planned to provide in the final analysis. Extrapolating on the results from the pilot analysis, we expect this will show methods that select similar feature sets and explain similarities in performance. We have added a paragraph describing this to the analysis section of the text:

Further analysis will look at the similarity between methods by considering the overlap in selected sets. This will help to explain method performance as we can relate differences in selected features to differences in performance and inspect which features top-performing methods select that others don't or if rather than selecting better features they exclude more uninformative features.

5. I think most benchmark datasets were likely annotated using the default feature selection methods in tools like Seurat or Scanpy. Therefore, it is important for authors to consider potential systemic bias within these benchmark datasets.

You are correct, thank you. The annotations we use as truth are the result of an analysis involving the processes we are trying to evaluate. It is unfortunately very difficult to avoid this bias for real datasets, but we are aware of it and will discuss it in the text. This is one motivation for including additional simulated datasets as suggested by other reviewers.

6. For the MILO metric, the author should give a clear explanation of why this method can be used as a metric for identifying unseen populations and should specify this metric calculated at the cellular level or at the category level. In addition, the author only used a subset of cells (up to 20,000 cells or 10 percent of the datasets, whichever is higher) and used five times the number of batches neighbors (up to a maximum of 200), do these changes have an impact on the evaluation of the method?

We based the MILO metric on how MILO was used to evaluate the identification of altered cell states by Dann et al. [11]. MILO is a differential abundance method that tests for changes in abundance in the neighbourhood of a cell. It first counts the number of cells in the neighbourhood coming from each sample and then tests for differences from what is expected considering a design matrix with variables of interest (in our case whether each sample is part of the query or reference). If neighbourhoods primarily consist of query cells, then they will be identified as significantly differentially expressed. Because we only consider cell labels that we know are not present in the reference by design, significant neighbourhoods in these populations indicate that variation present in the query but not the reference has been preserved. The statistical test is performed for each cell neighbourhood, which we aggregate the results by taking the proportion of cells with FDR less than 0.1 for each unseen cell label, and then by taking the average of these proportions across all unseen labels to get the final metric score. We have expanded on this in the text:

The results of the MILO test indicate whether query cells are enriched in a cell neighbourhood. We consider this a positive result for unseen cells as variation present in only the query has been conserved. The individual test results are summarised for each by taking the proportion of cell neighbourhoods significantly associated with the query (FDR adjusted p-value less than 0.1). The final overall score is the average of the proportions across all unseen labels.

The parameters you mention are used by the Dann et al. study and are defaults in the milopy package (the default is 10% of cells, we added the 20,000 cell floor). They are primarily chosen to reduce computational time. As they are suggested by the method authors we trust they will not significantly affect the results. When subsetting, we set a random seed so the same cells will be used for every MILO measurement for a dataset.

7. The tumor microenvironment (TME) exhibits apparent heterogeneity in different types of cancer. I recommend that authors should utilize multiple datasets about tumor TME to avoid possible evaluation bias from a single dataset.

The focus of the study is not cancer/TME but integration in general. We do not focus on cancer data specifically and believe it would be out of scope to do so. We may have caused confusion by the inclusion of the Reed breast dataset. Although we separate reference and query samples here based on BRCA status, all the samples in this dataset come from healthy individuals and do not include any tumour samples. We have made this clearer in the text.

Minor comments:

1. “Many computational scientists have tackled the integration problem using a variety of methods and there are now at least 200 tools for single-cell integration available”, which cited Ref2, but Ref2 is an article published in 2018. However, most integration methods for single-cell data were developed after 2018, so the authors should check the cited references.

The number cited in the manuscript comes from the scRNA-tools database (<https://www.scrna-tools.org/>) which was published in 2018 but has been continuously updated since. This database was the source for similar numbers in previous benchmarks and reviews and we believe this is the most up-to-date number available.

2. For the metric cell distance in mapping quality, how do you convert distance into the p-value of the chi-squared distribution, and what is the rationale behind this approach? And I think the metric cell distance in mapping quality is redundant.

- There is a formula/function for this, check details
- Needed a bounded number for metric score
 - May be alternatives...
- Check the second point but I think these are different things
 - Results of metric selection may show

The Mahalanobis distance can be described using a Chi-squared distribution with degrees of freedom equal to the number of variables (in this case the number of dimensions in the integrated embedding). Using this relationship a p-value can be calculated from the calculated distance with an appropriate CDF. The reason for using the p-value is that it is bounded between 0 and 1, unlike the distance which is unbounded. The description of this has been expanded in the text.

The Mahalanobis distance is naturally unbounded, so to get a score in the range 0 to 1 we convert it to a p-value using the relationship between the distance and a chi-squared distribution with degrees of freedom equal to the size of the integrated embedding.

Regarding the mapping quality cell distance metric we assume you feel it is redundant with the label distance metric. These are slightly different in that the cell distance measures the distance between query cells and reference label centroids while the label distance measures distances between query label centroids and reference label centroids. It is possible they measure the same information, if that is the case only one will be selected during the metric selection step.

3. All metrics the author used are normalized to [0,1], but why do some methods have negative values in their metrics?

Scores outside the range [0, 1] in the pilot data are the result of the scaling process. An effective range for each metric is established using a set of baseline methods and it is rescaled to this range. If another method performs better/worse than the baseline methods, it will get a value outside [0, 1]. We believe this is a useful feature of the scaling procedure as it provides context about how a method performs relative to the baselines.

4. There was an article on the evaluation of tools for highly variable gene discovery from single-cell RNA-seq data (PMID: 29481632), but the authors missed it.

Thank you for bringing that oversight to our attention. We have now cited that publication.

References

1. Luecken MD, Theis FJ. Current best practices in single-cell RNA-seq analysis: a tutorial. *Mol Syst Biol* [Internet]. 2019 [cited 2019 Jun 20];15. Available from: <https://www.embopress.org/doi/full/10.15252/msb.20188746>
2. Luecken MD, Büttner M, Chaichoompu K, Danese A, Interlandi M, Mueller MF, et al. Benchmarking atlas-level data integration in single-cell genomics. *Nat Methods* [Internet]. 2021; Available from: <http://dx.doi.org/10.1038/s41592-021-01336-8>
3. Zappia L, Phipson B, Oshlack A. Splatter: simulation of single-cell RNA sequencing data. *Genome Biol* [Internet]. 2017;18:174. Available from: <https://doi.org/10.1186/s13059-017-1305-0>
4. Lause J, Berens P, Kobak D. Analytic Pearson residuals for normalization of single-cell RNA-seq UMI data. *Genome Biol* [Internet]. 2021;22:258. Available from: <http://dx.doi.org/10.1186/s13059-021-02451-7>
5. Townes FW, Hicks SC, Aryee MJ, Irizarry RA. Feature selection and dimension reduction for single-cell RNA-Seq based on a multinomial model. *Genome Biol* [Internet]. 2019;20:295. Available from: <http://dx.doi.org/10.1186/s13059-019-1861-6>
6. Korsunsky I, Millard N, Fan J, Slowikowski K, Zhang F, Wei K, et al. Fast, sensitive and accurate integration of single-cell data with Harmony. *Nat Methods* [Internet]. 2019; Available from: <https://doi.org/10.1038/s41592-019-0619-0>
7. Lütge A, Zypych-Walczak J, Brykczynska Kunzmann U, Crowell HL, Calini D, Malhotra D, et al. CellMixS: quantifying and visualizing batch effects in single-cell RNA-seq data. *Life Sci Alliance* [Internet]. 2021;4. Available from: <http://dx.doi.org/10.26508/lsa.202001004>
8. Tran HTN, Ang KS, Chevrier M, Zhang X, Lee NYS, Goh M, et al. A benchmark of batch-effect correction methods for single-cell RNA sequencing data. *Genome Biol* [Internet]. 2020;21:12. Available from: <http://dx.doi.org/10.1186/s13059-019-1850-9>
9. Chazarra-Gil R, van Dongen S, Kiselev VY, Hemberg M. Flexible comparison of batch correction methods for single-cell RNA-seq using BatchBench. *Nucleic Acids Res* [Internet]. 2021;49:e42. Available from: <http://dx.doi.org/10.1093/nar/gkab004>
10. Kang JB, Nathan A, Weinand K, Zhang F, Millard N, Rumker L, et al. Efficient and precise single-cell reference atlas mapping with Symphony. *Nat Commun* [Internet]. 2021;12:5890. Available from: <http://dx.doi.org/10.1038/s41467-021-25957-x>
11. Dann E, Teichmann SA, Marioni JC. Precise identification of cell states altered in disease with healthy single-cell references [Internet]. *bioRxiv*. 2022 [cited 2022 Nov 21]. p. 2022.11.10.515939. Available from: <https://www.biorxiv.org/content/10.1101/2022.11.10.515939v1?ct=>

Response to reviewers: “Feature selection methods affect the performance of scRNA-seq data integration and querying” (NMETH-RR51186C)

Original reviewer comments (black), point-by-point responses (green), sections of the text (blue italic)

Reviewer 1

None

Reviewer 2

The authors have stuck to their registered analysis plan, and have not chosen to focus overly on any additional exploratory analyses. The analysis of batch-aware methods is an addition from the original analysis plan, and is well justified.

Overall, the authors have stuck closely to their registered analysis and executed it well - the presentation is generally very clear, and the authors should be commended for undertaking this important work for the single-cell research community. Figure 3 is especially clear, and informative. It provides a strong justification and important context for users when choosing a feature selection method.

Thank you for your kind appreciation of our approach and research plan. We also thank you for your kind words about the clarity of the results as we worked hard on how to clearly present a complex benchmark.

I have some minor comments regarding presentation of results, and suggestions to improve clarity.

In the second paragraph of the results "Metric selection is key for reliable benchmarking", it would help the reader if the authors could state explicitly what an ideal metric would look like, i.e. uncorrelated with number of features so as not to be biased by methods that select fewer or more features, has a dynamic range that allows for informative selection of methods, etc. This is implied by reading through the excluded metrics in table 1, but if stated up-front then it would be clearer.

Thank you for the suggestion. We have added the following sentence making this explicit.

“An ideal metric would accurately measure what it is designed for, returning scores across its whole output range that are independent of technical features of the data and are orthogonal to other metrics in the study.”

Expressing the standard deviation as the size of the square across figures makes it very hard to discern just how variable the correlations are. More so, because the size of the square is inversely proportional to the variation. This is likewise confounded with the colour range in Figure 1B and Figure 3B/C, in that white is in the middle of the range (correlations ~ 0), and so has a poor contrast with the light coloured background of the figure panel. I would suggest a supplementary figure that displays the variance of the correlations as a single value - this would significantly aid clarity of presentation. This is especially problematic because more variable methods actually have smaller points, which is counter-intuitive. I understand that the authors might want to emphasis the methods with a low correlation and low variance, but this current visualisation doesn't quite hit the mark.

Developing a visualisation for summarising so many data points while also showing variation was a difficult task, and while we think the final figures are effective, they have some shortcomings. As you suggested, we chose to reduce the size of highly variable data points to de-emphasise those values. Following your suggestion, we have created corresponding extended data figures for each main figure that uses this visualisation which show all data points that were summarised. These allow the reader to more clearly see both mean and variation using more space than was possible in the main figures. As an example, here is the corresponding extended figure for Figure 1B.

At the end of the penultimate paragraph of the section "The number of selected features affects performance", the authors state that "...this trend holds across all data sets, with more features required to achieve high scores...". This suggests that cell type diversity is roughly proportional to the number of features required to get a good integration and query-reference mapping. Could the authors discuss briefly at what point this could saturate, if at all? (modulo including all genes).

We don't believe the data from this study is sufficient to properly answer this question but as you suggest, it is likely that the ideal number of features is related to the biological and technical diversity of the dataset. We have added the following text to this paragraph to emphasise this point.

"However, the performance of selecting all features shows a limit to how much additional signal can be obtained. The number of features at which the additional signal saturates is unclear and is likely to be different for each dataset as a function of the biological and technical diversity that is present."

We also suggest this as an area of future research in the relevant paragraph of the discussion.

"Developing methods for automatically tuning the number of selected features based on technical aspects of datasets is a potential avenue for future research."

The text regarding the overall results is a little confusing (page 10). This is because the ordering in Figure 4A is not by highest overall ranking, but by the average overall score. So is Wilcoxon or

Seurat-VST the highest ranked method in this benchmark? At the moment the text and figure are not 100% harmonious. Perhaps this discrepancy comes from the 'Mean ranks' panel of Fig4A to declare the "winning method" rather than the "Overall score" which was used to order the methods in the figure. This requires some harmonisation to be 100% transparent and clear.

Thank you for your comment. Re-reading this section we agree that it is somewhat unclear. Several of the methods show very similar overall performance which we believe makes the ranking unstable as minor differences in scores can cause methods to be ranked several places higher or lower. The exception to this is Wilcoxon which, while it has the highest overall score, is clearly more variable across datasets. We have revised this paragraph for clarity and to place more emphasis on scores over ranks.

“Figure 4 shows the overall results for each metric category, sorted by the mean overall score across datasets for scVI integration (Extended Data Figure 5a). Several methods obtain similar average overall scores. The Wilcoxon method, the only method to select features using cell labels, has the highest average overall score but is more variable across datasets than other top-performing methods. This higher variability suggests that supervised selection of features may not be effective for all datasets, even when the same labels are used for evaluation, and that tuning the number of features selected using this approach could be required. The Seurat-VST method obtains the highest overall ranking and several other highly variable feature selection methods also perform well with similar mean scores and more consistent performance than Wilcoxon. The other top-performing alternative method is triku, which has similar overall scores to the highly variable selection methods but shows some bias towards batch correction over conserving biological variation.”

Page 34 - the sentence "...feature sets (have an insufficient dynamic range), were overly correlated with the number of features..." - could the authors define concretely what value or range of values they considered "overly correlated"?

Metrics were excluded by considering all the aspects we examined. For correlation with the number of selected features, we look at the values across all metrics to identify those that were unusually high rather than defining a specific threshold. The metrics excluded based on this criterion were kNN correlation and local structure which both showed consistently high correlations with a mean greater than 0.5 while the next highest metric had a mean of less than 0.4.

Reviewer 3

I am satisfied that the authors completed the experimental plan as described, exploratory analyses are appropriately described and caveated and the conclusions are justified. The manuscript and some figures have various typos/misspellings/missing words and would benefit from a careful edit.

Thank you for your confidence in how we completed the study. We have thoroughly proofread and edited the manuscript and hope we have corrected all small text errors.

Reviewer 4

Although this version has improved a lot, I still have the following minor comments:

Thank you for the appreciation. We have addressed the minor comments below.

1. Page 2/41: Reference 2 is from 2018, and the authors cite the integration of 250 tools (the previous version mentioned 200), which seems somewhat inappropriate.

As we discussed in responses to previous reviews, this reference is for a database recording tools for analysing scRNA-seq data that, while originally published in 2018 was continuously updated until May 2024 and currently contains 286 tools labelled with "Integration" (ie data integration). We do not claim to have investigated all of those tools but simply that they exist and consider it appropriate to cite the source of the number quoted in the manuscript.

2. Page 3/41: "avoid include several metrics which provide the same information" should be replaced with 'avoid including multiple metrics that provide the same information.'

Thank you for pointing this out, it has been corrected.

3. Page 7/41: "averaged over five sets" does not clearly explain what it represents. Additionally, although the authors mentioned four methods, they did not explain how the normalization was performed. The authors should use strict mathematical formulas to express this.

We have adjusted the text quoted to clarify that for the random feature sets metric scores were averaged over 5 random sets. We made sure that the scaling procedure is explained in Figure 2B and the relevant text with an accompanying formula.

4. Page 24/41: 'less than 10 percent of cells within that label, expressed in more than 80 percent of cells outside the label or with a p-value above 0.1. Next, the remaining features are sorted by estimated log-fold change, and the top 200 features are selected. The final feature set is the intersection of the features selected for each label.' Are the thresholds set too arbitrarily, such as $p > 0.1$ and the top 200?" Do different parameter settings affect the results

These thresholds were set based on our experience of common practice for defining marker genes for scRNA-seq populations. Using different thresholds may affect results but we did explore alternatives as our focus was not on supervised methods as they can often not be used in the setting we are interested in (i.e. cell labels are often not available prior to integration as mentioned in the text). We have expanded the relevant part of the discussion to make this point clearer.

“Label-guided marker genes (Wilcoxon) also performed very well but were more variable across datasets. We focused on unsupervised methods and other supervised techniques may produce more stable results. However, supervised feature selection only applies when cell labels are available, typically not the case before integration.”